# Genome-level selection in tumors as a universal marker of resistance to therapy

Erez Persi [1] ✉, Praneeth R. Sudalagunta [2], Yuri I. Wolf [1], Rafael R. Canevarolo [2], Mehdi Damaghi [3], Kenneth H. Shain[4], Ariosto S. Silva [2] ✉ & Eugene V. Koonin [1] ✉

Tumor evolution is shaped by selective pressures imposed by physiological factors as the tumor naturally progresses to colonize local and distant tissues, as well as by therapy. However, the distinction between these two types of pressures and their impact on tumor evolution remain elusive, mainly, due to extensive intra-tumor heterogeneity. To disentangle the effects of these selective pressures, we analyze data from diverse cohorts of patients, of both treated and untreated cancers. We find that, despite the wide variation across patients, the selection strength on tumor genomes in individual patients is stable and largely unaffected by tumor progression in the primary settings, with some cancer-specific signatures detectable in the progression to metastases. However, we identify a nearly universal shift toward neutral evolution in tumors that resist treatment and demonstrate that this regime is associated with worse prognosis. We validate these findings on both published and original datasets. We suggest that monitoring the selection regime during cancer treatment can assist clinical decision-making in many cases.

Tumor evolution is initiated by oncogenic driver mutations that transform a normal cell to a neoplastic state, conferring selective advantage and growth potential[1,2]. This evolution is underpinned by genome instability and the emergence of genetically heterogenous subclones, which translates into phenotypic heterogeneity[3–8]. The widespread intra-tumor heterogeneity enhances the ability of tumors to resist selective pressures imposed by physiological barriers, such as immune response and colonization of metastatic niches, or treatment. Eventually, tumor evolution results in outgrowth of resistant clones, due to either pre-existing or treatment-induced mutations, aggressive tumors, and inferior clinical outcomes[9–18]. Therefore, it is critical to understand how different selective pressures shape the clinical course of tumor evolution, and the mechanisms behind treatment resistance[19–23]. This can be partially achieved by analysis of tumor biopsies from patients with temporal or multi-region sequencing, allowing phylogenetic reconstruction of tumor evolution in a patient[24–26].

Generally, in untreated cancers, little heterogeneity is observed with respect to point mutations in known driver genes, suggesting early metastatic seeding[27], whereas treatment has been found to primarily promote selection of pre-existing resistant subclones, reducing clonal diversity[28]. However, beyond these general trends, there are also substantial patient- and cancer-specific differences. For example, in breast cancer, early divergence of metastases is common, but in many patients, metastases appear to be seeded from different clones in the primary tumor, in the absence[29,30] or in the presence[31] of therapy. Under therapy, the mutational landscape is even more complex, with metastases typically exhibiting substantially higher mutational burden and clonal diversity compared with early breast cancer[32–34]. About 25% of recurrent invasive breast tumors are clonally unrelated to their respective pre-invasive early ductal carcinoma in situ[35]. Further, many patients acquire drug-resistant mutations following neoadjuvant targeted therapy, manifesting drastic selective sweeps and outgrowth of treatment-induced resistant subclones[36–38]. Evidence of the role of

[1]Computational Biology Branch, Division of Intramural Research, National Library of Medicine, National Institutes of Health, Bethesda, MD, USA. [2]Department of Metabolism and Physiology, H. Lee Moffitt Cancer Center and Research Institute, Tampa, FL, USA. [3]Department of Pathology, School of Medicine, Stony Brook University, Stony Brook, NY, USA. [4]Departments of Malignant Hematology and Molecular Medicine, H. Lee Moffitt Cancer Center and Research Institute, Tampa, FL, USA. ✉e-mail: erezpersi@gmail.com; Ariosto.Silva@moffitt.org; koonin@ncbi.nlm.nih.gov

treatment-induced mutations as drivers of drug-resistant metastases exists for many cancers and diverse therapies[39,40]. The effect of treatment goes beyond the tumor itself, as exemplified by clonal replacement of tumor-specific T-cells under immunotherapy[41].

Similarly, complex mutational landscapes that translate into diverse phylogenetic patterns occur to different degrees across cancer types. In untreated cancers, as exemplified by diverse metastatic dissemination patterns and modes of evolution (e.g., linear, branching, and punctuated)[42–44], as well as heterogeneity among regional primary samples[45]. In treated cancers, the diversity can manifest as large clonal shifts and heterogeneous clonal architectures, often enriched with treatment-induced mutations, under diverse modes of evolution (e.g., divergent or convergent)[46–54]. For example, in lung cancer patients, convergent evolution and selective sweeps have been observed in the early evolution of lung tumors[55], followed by diverse modes of evolution towards invasiveness (with or without link to the pre-invasive tumor)[56], and an increase in the genetic diversity under chemotherapy, which triggers relapse[57]. Further, heterogeneity is context-dependent and varies between cancer subtypes, as exemplified in Melanoma, where low intra-tumor heterogeneity is generally observed among lesions under treatment[58,59], but not in Uveal Melanoma, a rare (3%) and aggressive form of the disease exhibiting increased heterogeneity[60]. These substantial differences and diversity among patients and cancer types emphasize the need for establishing additional metrics that correlate with clinical outcomes to assist decision making in the clinics, in particular establishing criteria to assess response to treatment[61,62].

To identify universal evolution-guided markers of response to treatment, we here extend analysis of cancer patients with multiple samples. We quantify the evolutionary state of an entire tumor genome by the overall mutational burden, defined as the number of non-synonymous, protein-changing mutations ($N$), and the strength of selection at the protein level, defined as the ratio between the rates of non-synonymous and synonymous ($S$) mutations, $dN/dS$. The values of $N$ and $dN/dS$ at the genome level are universal parameters that characterize the course of tumor evolution and strongly correlate with patients' clinical outcomes[5,63]. Recently, this predictive capacity has been further demonstrated following immunotherapy[64,65]. Under this parametrization, tumor evolution is predominantly neutral ($dN/dS \approx 1$) but is also non-monotonic. In initial phases (low $N$), tumors accumulate a few positively selected driver mutations ($dN/dS > 1$) and the tumor fitness increases (worse prognosis) with $N$. However, as the tumor progresses, accumulation of many (slightly) deleterious passenger mutations becomes a burden to the tumor genome (high $N$) and the tumor fitness decreases (improved prognosis) with $N$, with detectable signatures of purifying selection ($dN/dS < 1$) that acts to remove these mutations. This non-monotonic, critical behavior, with the highest tumor fitness around neutrality ($dN/dS \approx 1$) and intermediate $N$, is consistent with theory[66], as well as with the observations that most of the somatic mutations in cancer are neutral[67]. Nevertheless, there are notable deviations from neutrality, with signatures of both positive and purifying selection observed at the gene level[7,68,69] and at the genome (patient) level[63,70–72]. However, most of these observations are based on the analysis of snapshots of a single primary sample from individuals in the TCGA database. Thus, it remains an open question how tumor natural progression and treatment separately shape this process, and if universal variations in $N$ or $dN/dS$ can be identified that might be valuable for clinical decision-making. Applying this approach to untreated and treated cancers, we demonstrate that generally, within a cohort of patients, there is a wide distribution of $dN/dS$ values around neutrality. In untreated cancers this distribution is roughly invariant in each patient, in the primary settings, with some cancer-specific signatures observed as tumors naturally progress to metastases. By contrast, in treated, resistant cancers, nearly universally, the

$dN/dS$ distribution shifts toward neutrality. We demonstrate that this bias towards neutrality correlates with inferior clinical outcomes and suggest how to exploit it for treatment guidance.

## Results

To disentangle the effects of therapy from the effects of natural tumor progression, we analyzed diverse datasets of both treated and untreated tumors of patients for which at least 2 samples were available, and evaluated the $dN/dS$ ratio from the whole-exome sequencing (WES) ("Methods"). This analysis allows us to determine how treatment and natural tumor progression separately affect the selection regimes of tumor genomes, and to identify potential variations associated with treatment failure. We start with the analysis of untreated cancers, followed by analysis of treated cancers, and finally validate our findings on both published and original datasets. In the following, 'quantifiable' patients refer to those with at least two samples available for comparison (e.g., different primary; pre- vs. post-treatment; primary vs. metastases), and valid $dN/dS$ estimates in both compared samples ("Methods").

### Selection in untreated cancers

To test the effect of natural progression of tumors, without therapy, we analyzed 3 published datasets of untreated cases, including 23 colorectal cancer patients[42], 39 esophageal cancer patients[43] and 40 small cell lung cancer patients[45] (Fig. 1 and Figs. S1-S2). These cohorts represent a diversity of phylogenetic patterns across patients. In the colorectal cohort, early metastatic dissemination is dominant[42], whereas in the esophageal cohort linear, punctuated, or divergent phyletic patterns are observed[43]. All cohorts include heterogeneity among regional primary tumors.

The 21 quantifiable patients in the colorectal cohort included untreated primary and metastatic samples from each patient, and multiple primary samples for 9 patients. Metastases include both synchronous (mainly from the LN and/or liver) and asynchronous distant samples (mainly from brain and lung). We excluded post-treatment asynchronous metastases samples from 6 patients, such that the analysis represents only untreated cases. In a patient, the average $dN/dS$ of primary samples did not differ significantly from the average value of the metastatic samples, that is, a roughly linear relationship (slope close to 1) was observed between the $dN/dS$ values of primary tumors and metastases, even when $N$ changed significantly (Fig. 1a and Fig. S1). A linear relationship was also observed in the comparison between early and late primary samples, that was statistically indistinguishable from the regression obtained in the progression to metastases (ANOVA, F-test) (Fig. 1a). In the esophageal cohort, multiple primary samples (at least 0.5 cm apart) were available for all 39 patients, and for 10 patients, there were also 1–3 LN samples. The analysis of this cohort also showed a wide distribution of $dN/dS$ values across patients along with a robust linear relationship between the $dN/dS$ values of samples in a patient, with natural progression towards LN metastases exhibiting a signature of positive selection (Fig. 1b and Fig. S2). In this case, positive selection manifests as higher $dN/dS$ values (and typically, greater than 1) in the LN compared with the primary samples, with a steep slope, whereas the relationship between early and late primary samples is close to linear ($P$ value < 0.018, ANOVA F-test) (Fig. 1b). In the lung cancer cohort, 3 primary tumors were collected from each patient. Comparing the $dN/dS$ values of 36 quantifiable patients for early and late primary samples, we observed a close to linear relationship and wide distribution of $dN/dS$ values across patients (Fig. 1c).

Overall, the analysis of these untreated cancer cases shows a wide range of $dN/dS$ values across patients in each cancer type, with stable $dN/dS$ values in each patient, that are largely unaffected by natural cancer progression in the primary settings. Nonetheless, natural progression toward metastases manifests cancer-specific signatures of

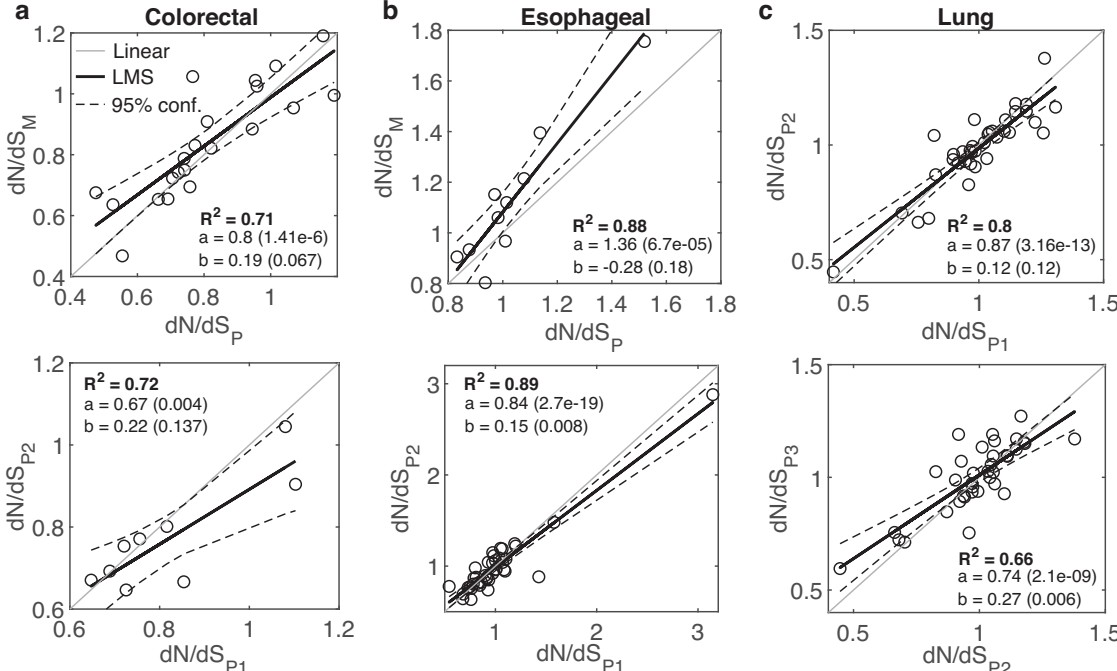

**Fig. 1 | Evolution regimes in Untreated Cancers. a** The relationship between $dN/dS$ of primary (P) samples and metastatic (M) samples, prior to treatment, in 21 colorectal cancer patients[42] (**top**), and the relationship between $dN/dS$ of early primary (P1) samples and late primary (P2) samples in 9 patients (**bottom**). **b** The relationship between $dN/dS$ of P samples and M samples, in 10 esophageal cancer patients[43] (**top**), and the relationship between $dN/dS$ of P1 samples and P2 samples in 9 patients (**bottom**). **c** The relationship between $dN/dS$ of 3 primary samples in 36 quantifiable lung cancer patients[45] are shown for sample 1 vs. sample 2 (**top**) and sample 2 vs. sample 3 (**bottom**). R-square ($R^2$) of a least mean square (LMS) fit, and the linear regression fit (y=ax+b) with respective P-values of the slope (**a**) and intersection (**b**) are shown. Confidence Intervals (95%) are denoted by dashed curves. See main text for the one-way ANOVA tests comparing between two linear regressions ("Methods").

positive (or relaxed) selection, consistent with the conclusions of previous pan-cancer analysis[28], and as exemplified by the esophageal cohort data, a cancer that was previously reported to exhibit signatures of positive selection[73].

## Selection in treated resistant cancers

To identify the impact of therapy on tumor evolution, we analyzed samples before and after treatment from 5 cohorts of treated cancers, where eventually all patients developed resistance to the applied therapy. These cohorts include patients with pediatric B-acute lymphoblastic leukemia (ALL)[46], chronic lymphocytic leukemia (CLL)[48], ER-positive breast cancer[38], urothelial carcinoma[52] and glioblastoma[54]. The changes in N following therapy vary from cohort to cohort, yet we find a systematic effect on $dN/dS$ as explained below (Fig. 2 and Fig. S3).

The ALL cohort included 16 (out of 20) quantifiable children who received daily standard chemotherapy (6MP; Mercaptopurine) for a year following diagnosis, and all cases recurred within several months (<36 m). The $dN/dS$ values before and after treatment significantly correlated, with a shift towards neutrality at relapse, whereby in cases with $dN/dS < 1$ before treatment, the $dN/dS$ values tended to increase at relapse, and in cases with $dN/dS > 1$, there was a decreasing trend of $dN/dS$ at relapse (Fig. 2a). This shift manifests as a statistically significantly shallower slope, compared to a linear reference model that represents the variance in the data ("Methods") ($P$ value < 0.057; ANOVA F-test). The post-treatment $dN/dS$ values close to neutrality were associated with the worst outcome (rapid relapse), suggesting that the shift to neutrality is associated with higher tumor fitness and treatment failure (Fig. S4).

The CLL cohort tracks 8 patients that previously received 1 to 8 lines of standard treatment, and then were treated with venetoclax (BCL2-inhibitor) but developed resistance with median time of 15 months to relapse. Four cases developed the aggressive Richter's transformation. Similarly to the ALL cohort, $dN/dS$ values of 7 quantifiable patients were highly correlated with a tendency to neutral evolution following treatment at relapse ($P$ value = 0; ANOVA F-test) (Fig. 2b). Further, patients whose pre-treatment tumors were further from neutrality (C811, C812), had the best response but eventually relapsed, concomitantly with their $dN/dS$ values approaching neutrality. Patients with stable $dN/dS$ values close to neutrality exhibited the worst response, whereas in cases that drifted away from neutrality (C586, C577), the time to relapse was longer (Fig. S5), consistent with the results in the ALL cohort.

The breast cancer cohort consisted of patients who developed resistance to ER-directed therapies (e.g., Aromatase inhibitors), with a signature of acquired HER2 mutations in the metastatic settings. Consistently, also in this cohort, the $dN/dS$ values in the 7 out of 8 quantifiable patients showed a tendency to neutrality in post-treatment metastatic samples (from distant organs) compared to the primary naïve samples (P-value = 0.008; ANOVA F-test) (Fig. 2c).

The bladder cancer cohort consisted of 16 quantifiable patients who developed resistance to chemotherapy. Along with a single treatment-naive primary sample, the post-treatment samples were available from the primary tumor in some patients, whereas in others, they were collected from different distant metastases (e.g., LN, prostate, liver, lung). The $dN/dS$ values before and after treatment were correlated, but most patients had values below 1, such that the tendency to neutrality post treatment was less pronounced than in other cohorts, although still apparent ($P$ value = 0.098; ANOVA F-test) (Fig. 2d). Three patients received previous Bacillus Calmette Guerin (BCG) intravesical immunotherapy, which might explain some outliers.

Last, the cohort for Glioblastoma, an aggressive brain tumor with limited therapeutic options, consisted of patients who received

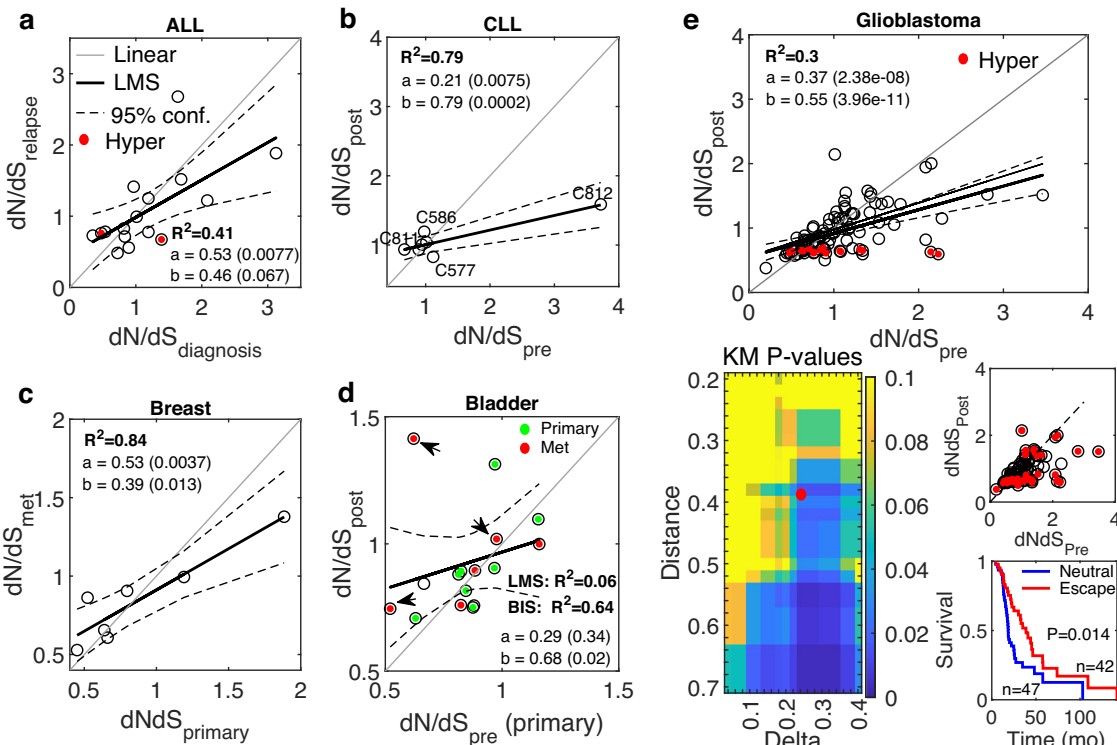

**Fig. 2 | Evolution regimes in resistant treated cancers. a** The relationship between *dN/dS* at diagnosis and after chemotherapy at relapse of 16 pediatric ALL patients[46]. Cases that developed a hypermutator genotype after treatment are marked (*red*). **b** The relationship between *dN/dS* before and after BCL-2 inhibition treatment of 7 CLL patients[48]. The cases C811, C812, C586, and C577 (see main text) are marked. **c** The relationship between *dN/dS* of treatment-naïve primary samples and post-treatment distant metastases, of 7 ER-positive breast cancer patients following ER-targeted therapy[38]. **d** The relationship between *dN/dS* of a treatment-naïve primary sample and the average *dN/dS* of primary (*green*) and metastases (*red*) samples post treatment, of 16 bladder cancer patients[52]. 3 cases that previously received BCG are marked (*arrow*). R² of a Bi-square (BIS) fit is also shown due to the outliers. **e** The relationship between *dN/dS* before and after radiotherapy of glioblastoma patients[54] (*top*). Cases that developed hypermutator genotypes

following treatment are marked (*red*). Linear fit curve is shown with (*thick*) and without (*thin*) the hypermutator cases. Heatmap of *P* values of the Kaplan-Meier survival analysis of glioblastoma patients, with respect to *dN/dS* of paired samples (before and after treatment), comparing patients in the neutral regime or approaching it after treatment with patients far-from or escaping the neutral regime (***bottom, left***). The point marked in red (Distance = 0.38, Delta = 0.24) serves an example to demonstrate patient's classification and the respective survival curves (***bottom, right***). R-square (R²) of a least mean square (LMS) fit, and the linear regression fit (y = ax+b) with respective *P* values of the slope (**a**) and intersection (**b**) are shown. Confidence Intervals (95%) are denoted by dashed curves. See main text for the one-way ANOVA tests comparing between the linear regressions of each cancer relative to a reference linear model ("Methods").

radiotherapy and alkylating agent temozolomide (TMZ). The *dN/dS* values of 89 (out of 114) quantifiable patients before and after treatment were highly correlated, with a clear tendency to neutrality in the post-treatment samples (*P* value = 0; ANOVA F-test) (Fig. 2e). Post-treatment, many patients developed hypermutation, which was associated with a clear decrease in *dN/dS* and purifying selection, presumably, to mitigate the accumulation of deleterious mutations. The shift to neutrality was apparent and significant regardless of the inclusion or exclusion of these hypermutator cases. To test the association of the shift to neutrality with tumor fitness post-treatment, we performed a systematic Kaplan-Meier survival analysis, comparing patients near neutrality or approaching the neutral regime post treatment with patients that are far from or escape the neutral regime ("Methods"). The results show that the post-treatment shift to neutral evolution was significantly associated with worse prognosis, with the survival of patients improving with the distance from neutrality (Fig. 2e). These findings suggest that the neutral regime post treatment correlates with increased tumor fitness, beyond the general association of neutrality with worse prognosis in this cancer that was captured in previous analysis of primary samples from the TCGA database[63].

Overall, like in the untreated cases, *dN/dS* values in these treated cancers were largely linearly correlated. However, as therapy fails and tumors become resistant, a systematic shift towards neutral evolution

was detected. This shift appears to be associated with inferior clinical outcomes, corroborating that neutral evolution corresponds to the highest tumor fitness[5]. Nonetheless, it should be stressed that the shift towards neutrality is not always apparent. For example, in cases characterized by extremely low intra-tumor heterogeneity, as previously reported for melanoma patients under chemotherapy[58], *dN/dS* values are expected to be highly conserved, with no apparent effect of therapy or natural progression. This is indeed the case for this cohort (Fig. S6). Thus, additional effort is required to identify the dynamics of *dN/dS* in a patient in such cases, as we demonstrate below.

**Validation on published datasets of breast cancer patients**
To validate the observations on the effects of natural progression and treatment on cancer evolution for one cancer type, we harnessed the diversity of breast cancer studies that included both treated and untreated patients, totaling 5 cohorts ("Methods" and Fig. S7). Analysis of these 5 cohorts allowed for comparisons between regional primary tumors[36,37], and between primary tumors and early local[29] or distant metastases[30,38], with and without therapy (Fig. 3).

The combined results from these 5 studies show that, in the untreated cases, the *dN/dS* values among primary tumors were strictly linearly correlated, consistent with the results on several types of untreated cancers presented above (*cf.* Fig. 1); however, when primary

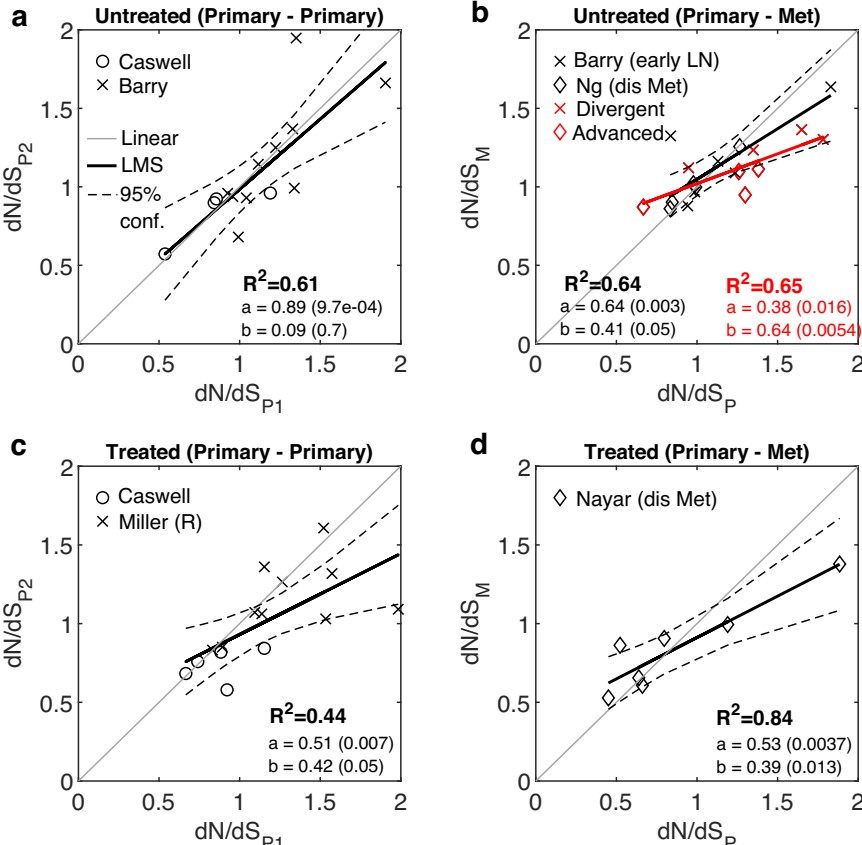

**Fig. 3 | Analysis of published datasets of untreated and treated breast cancers.** **a** The relationship between $dN/dS$ values of early and late primary untreated samples, combined from two studies[29,37]. **b** The relationship between $dN/dS$ values of primary and metastatic untreated samples. Metastatic samples are comprised of only local LN (without distant metastases) in one study[29], or distant metastases in another study[30]. Divergent cases in the former and advanced cases in the latter and their fit are displayed (red). Fit without these cases is also shown (black). See also Fig. S8. **c** The relationship between $dN/dS$ values of primary tumors before and after treatment, combined from two studies[36,37]. The cases treated with HER-positive directed therapy from[37] and the resistant (R) tumors from[36] are shown. **d** The case of resistant breast tumors to ER-directed agent comparing primary and metastases (*cf*. Fig. 2c)[38], is re-plotted for convenience. R-square ($R^2$) of a least mean square (LMS) fit, and the linear regression fit (y = ax+b) with respective *P* values of the slope (a) and intersection (b) are shown. Confidence Intervals (95%) are denoted by dashed curves. See main text for the one-way ANOVA tests comparing between the linear regressions of each cancer relative to a reference linear model ("Methods").

tumors were compared to metastatic samples, a shift to neutrality was revealed even in the absence of therapy (*P* value = 0.052; ANOVA F-test) (Fig. 3a, b). These findings were consistent with the notion of cancer-specific signatures of relaxed selection, in the natural progression to metastasis, derived from the analysis of other cohorts (see above). Considering the clinical information on the patients, the shift to neutrality appeared to originate from either early divergence of local LN in one study[29] or advanced stages (T4b, T4d) in another study[30] (Fig. 3b and Fig. S8). All the other cases, from both studies, appeared strictly linear, with only one outlier exhibiting a strong APOBEC-driven mutagenesis and high $dN/dS$ (Fig. S8), consistent with signatures of positive selection in such cases[74,75]. Further, patients under treatment who developed resistance exhibited a clear tendency to neutral evolution post-treatment, both in the primary tumors (*P* value = 0.044; ANOVA F-test) (Fig. 3c) and in the progression to metastatic tumors (*P* value = 0.008; ANOVA F-test) (Fig. 3d), in agreement with the observations on other cancers (*cf*. Fig. 2).

Overall, the analysis of these breast cancer cohorts presents a more complicated picture, whereby natural progression to advanced stages can also be associated with a shift toward neutrality, as a form of weakened selection, specifically, in early divergence and advanced stages, but also that treatment can cause a shift to neutrality even in the primary settings. These findings emphasize the need for further validation of the effects of therapy, for which longitudinal and temporal sampling of plasma cell cancers could be adequate[15] because in these cases, it is easier to monitor the effects of therapy along the disease progression in the primary settings.

## Validation on an original dataset from multiple myeloma patients

To test and validate the effect of therapy, we analyzed an original cohort of 624 multiple myeloma patients with 780 bone marrow aspirates, such that sequential biopsies across multiple stages were available from more than 100 patients (Fig. 4 and Fig. S9). The cohort spanned all stages, from very early, pre-malignant "benign" ones, including the monoclonal gammopathy of undetermined significance (MGUS) and the Smoldering multiple myeloma, followed by newly diagnosed treatment-naïve active disease ('Pre' state), to the early and late recurrences of the disease under continuous treatment ("Methods"). We assessed individual clinical response according to a consensus of measurements, including kappa/lambda light chains ratio (K/L) and serum M-spike from standard of care laboratory results ("Methods"). 92% of the pairs of sequential biopsies consisted of two therapy refractory (64%) or a first therapy sensitive biopsy followed by therapy resistant one (28%), highlighting the prominence of emerge of therapy resistance in MM (Fig S10). Thus, like in other treated cancers above, resistance to treatment was paramount.

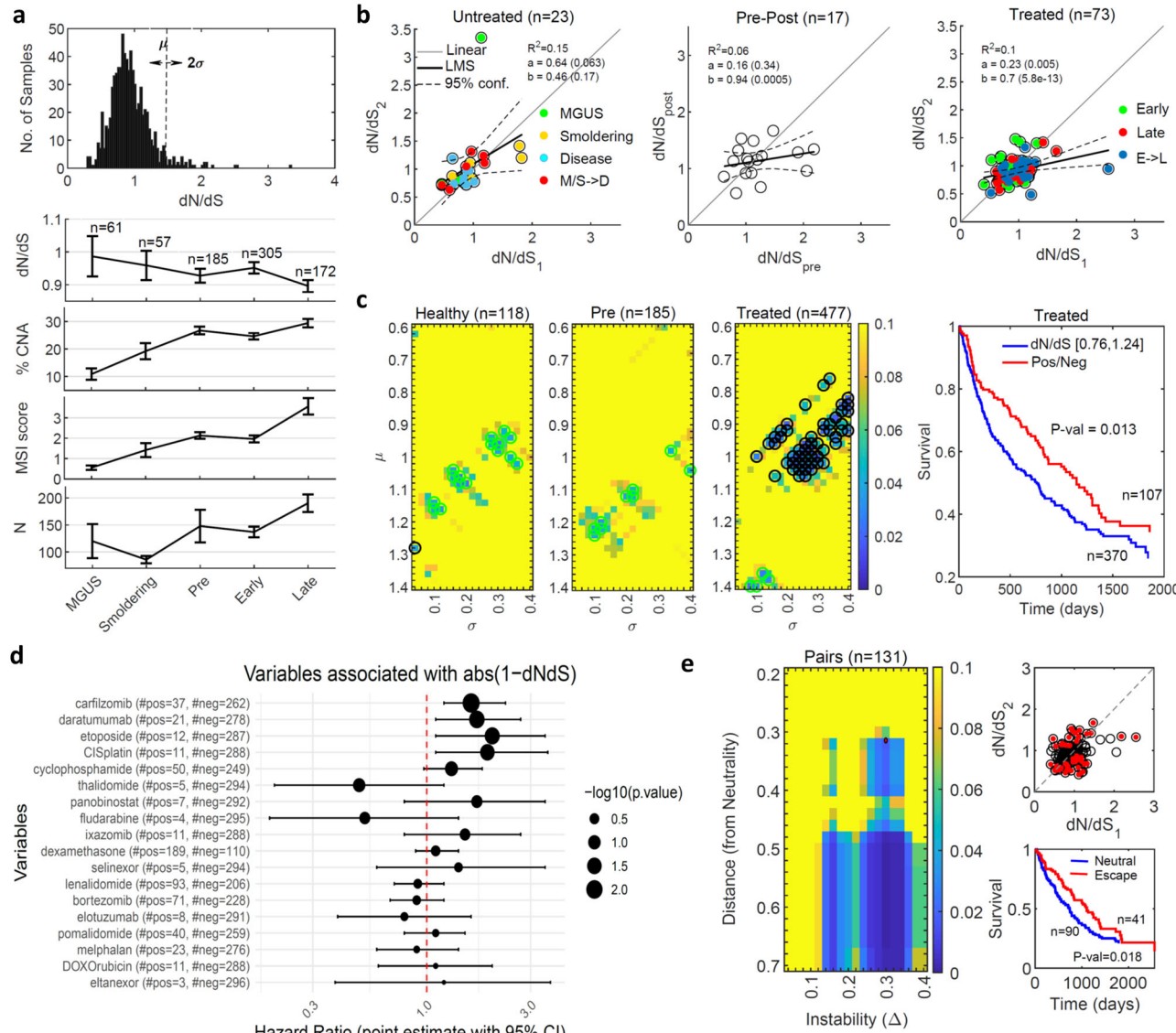

**Fig. 4 | Analysis of the multiple myeloma patients cohort. a** The distribution of *dN/dS* values across 780 samples (***upper***). The median of *dN/dS* values, percentage of genes affected by CNA, MSI score and the no. of *N* mutations across different statuses of the disease progression, error bars denote standard deviation of the mean (SEM) (***bottom***). **b** The relationship between *dNdS* values of early (1) and late (2) samples of the same patient are shown for 23 patients with untreated samples, including the healthy benign states MGUS and Smoldering and the disease state at diagnosis (***left***), 17 patients with disease state before (*pre*) and after (*post*) treatment (***middle***), and 73 patients who experienced disease relapse under continuous treatment (***right***). Colors code denotes comparisons of the same status, and when the status changes (arrow). The results of linear regression fits are denoted. One-way ANOVA F-test comparing untreated and treated linear regressions provides *P* value < 0.073 (see main text). **c** Heatmap of the *P* values of Kaplan-Meier (KM) survival analysis with respect to *dN/dS*, comparing values in the range μ±σ (as

shown in panel A) with values outside this range, of samples in the healthy statuses (i.e., MGUS and smoldering) prior to treatment (***left***), in the disease status at diagnosis prior to treatment (***middle***) and in the relapse status (i.e., early and late) under treatment (***right***). Circles correspond to *P* values < 0.05, where cyan circles indicate better prognosis (valleys in tumor fitness) and black circles indicate worse prognosis (hills in tumor fitness) of samples in the chose range. An example of the actual KM curves for the choice μ = 1, σ = 0.24 is shown on the right. **d** Cox regression analysis of different drugs with respect to the distance from neutrality post treatment, |*1-dN/dS*|, indicating that most of the drugs lead to regime close to neutrality (HR > 1). **e** Heatmap of P-values of KM analysis for paired samples before and after lines of therapy, comparing patients in the neutral regime or approaching it after treatment with patients far-from or escaping the neutral regime (***left***). The point marked in red (Distance = 0.32, Delta = 0.3) serves an example to demonstrate patient's classification and the respective survival curves (***right***).

Analysis of this cohort reveals *dN/dS* values across samples that are distributed close to neutrality, whereby disease progression correlated with an increase in the number of *N* mutations, MSI and CNA, and with a decrease in *dN/dS* (*dN/dS* < 1), particularly, in late relapsed refractory stages (Fig. 4a). Presumably, the decreased *dN/dS* is a consequence of balancing the accumulation of deleterious passenger mutations. No significant differences between stable and unstable genomes were detected in the distribution of *dN/dS* values

("Methods"). Notably, in the pre-treatment cases, there was a close to linear relationship between the *dN/dS* values, irrespective of the progression, that is, between consecutive samples of the same stage, as well as under progression from MGUS and Smoldering multiple myeloma to the treatment-naïve active disease (Fig. 4b). Comparison of pre- and post-treatment sequential biopsies confirmed a break in the correlation and a shift towards neutrality post-treatment, which persisted through early and late relapsed refractory disease (Fig. 4b). The

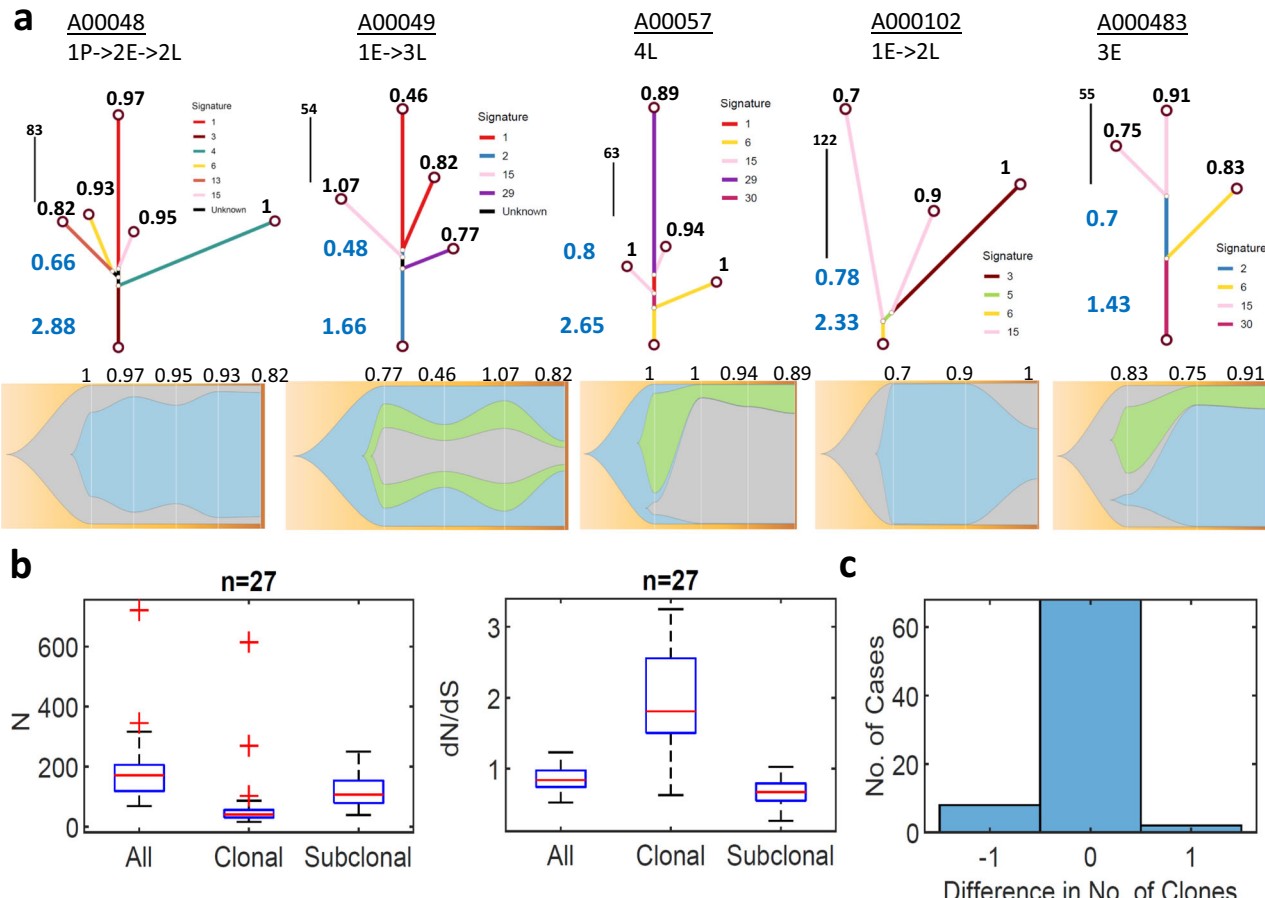

**Fig. 5 | Analysis of single patients with multiple samples in the multiple myeloma cohort. a** Representative examples of phylogenetic trees of patients with multiple samples; trajectories are denoted as arrows from pre-treatment (P) to early (E) and late (L) recurrences of sequential biopsies (***upper***). Dominant COSMIC mutational signatures along the tree branches (color coded), mirroring the dominant signatures in the entire cohort (see Fig S12). *dN/dS* values of trunk and branches (blue) and of each leaf (black) are denoted. The respective clonal compositions, of sequential biopsies, are shown for each patient, with the *dN/dS* of each sample (***bottom***). **b** The distributions of N mutations and *dN/dS* values across patients with at least 3 samples (n = 27), as deduced from the phylogenetic trees of

each patient, for all mutations, clonal (trunk, shared by all) mutations and subclonal (branch, private) mutations. Box plots denote the median (red) and the edges of the box are the 25th and 75th percentiles, with whiskers extending to the most extreme data points. See Fig. S13 for similar analysis in other cohorts, indicating that positive selection in the trunk and negative selection at the branches is a universal feature. **c** Histogram of the difference in the no. of clones between consecutive biopsies under treatment in a patient, indicating that the no. of clones is highly stable (but with dynamic proportions as in panel A), with a weak tendency for reduced clonality.

shift towards neutrality from the natural progression in the absence of therapy to progression under therapy was apparent (*P* value < 0.073, ANOVA F-test).

To test the association of this shift with the clinical outcome of the patients, we performed a systematic Kaplan-Meier survival analysis with respect to the *dN/dS* values. Specifically, we compared all possible groups of patients in a range of *dN/dS* values (μ ± σ) to those outside this range. We found that deviations from neutrality post treatment were associated with better prognosis (i.e., valleys in the tumor fitness landscape), whereas under treatment the shift to neutrality was pronounced and associated with worse prognosis (i.e., hills in the tumor fitness landscape) (Fig. 4c). Additional tests further substantiated the association of this dynamics towards neutrality with poor clinical outcome. Survival tests comparing different regimes of positive against negative selection did not yield any significant correlation (Fig. S11), confirming that deviation from neutrality, either positive or negative, have equivalent positive prognostic value. Patient tumors sensitive to therapy (i.e., measurable clinical response), showed evidence of deviation from neutrality (Fig. S10), consistent with the observations on other cancers (*cf*. Figs. S4, S5). Cox proportional hazard analysis of the effect of different treatments on the dynamics of

*dN/dS* confirmed that most drugs shifted *dN/dS* values closer to the neutral regime (Fig. 4d). Last, we performed a systematic survival analysis of paired samples, before and after treatment ("Methods"), similarly to the case of glioblastoma (*cf*. Fig. 2e). Patients classified as far from or escaping the neutral regime after treatment had better prognosis, whereas those near the neutral regime or approaching it after treatment had worse prognosis (Fig. 4e). This difference was more pronounced with the increasing distance from neutrality (Fig. 4e). Thus, similarly to the case of glioblastoma, disease progression under resistance to treatment correlated with neutrality and worse prognosis.

To better understand the dynamics of *dN/dS* throughout the evolution of tumors in individual patients, we reconstructed phylogeny, along with mutational signatures and clonal composition of each sample ("Methods"), as exemplified on several representative patients (Fig. 5a). The mutational signatures along the trees indicate that beyond COSMIC signatures 1 and 2 (that appear in most cancers), the additional significant signatures were associated with DNA damage (COSMIC signatures 3,6 and 15), and this was the case in the entire cohort (Fig. S12). Notably, *dN/dS* values for shared mutations at the trunk of each tree (clonal) were consistently larger than 1, clearly

indicating positive selection at the onset of tumor evolution. In contrast, dN/dS values for branch mutations (subclonal enriched) were significantly below 1, indicating negative selection (presumably, of deleterious passenger mutations) in late stages. When all mutations were considered, these opposing trends largely canceled out, resulting in an effectively neutral evolutionary regime in late stages (leaves). Testing this bi-phasic tumor evolution systematically for all patients in this cohort (Fig. 5b), as well as in other cancers (Fig. S13), unequivocally demonstrated that that this pattern was universal and independent of the shape of trees or treatment status. The strong negative selection at the branches could be thought to translate into reduced clonality. However, analysis of the clonal composition indicates that the number of clones is small and highly stable (typically 2–3), with only a few cases manifesting reduced clonality (Fig. 5c). Thus, selection mostly affects the proportional size of clones, not their number. This suggests that non-genetic changes, such as epigenetic modifications, may facilitate the diversity among cells of each clone to increase (or maintain) tumor fitness, under selective pressures ("Discussion").

Overall, these results, from individuals with sequential biopsies further validate and corroborate the expectations from theory and previous empirical evidence of the non-monotonic behavior of dN/dS during tumor progression. At initial phases (typically, characterized by low N), dN/dS increases with N due to accumulation of driver mutations, whereas at later stages (at high N), dN/dS levels off or even decreases with N due to the accumulation of deleterious passenger mutations that become a burden to the tumor and are eliminated by purifying selection[5,63,66].

## Analysis of the relationship between dN/dS and tumor size

To gain further insight into the nature of the neutral regime, we analyzed the relationship between dN/dS and the respective tumor size across studies. This is because, theoretically, the neutral evolution regime following therapy could result from shrinkage of the tumors, leading to a population bottleneck and enhanced genetic drift[76,77]. However, addressing this possibility by comparing the dN/dS values to the tumor size (in cases where data was available), did not provide supporting evidence (Fig. S14). Generally, there was no significant correlation between these two variables although, in several studies, neutrality was associated with large tumor size, and in only one study, we found a weak link between small tumors and neutrality (Fig. S14). Thus, the shift to neutrality appears to be associated primarily with tumor growth, rather than with population bottlenecks and drift although the latter might be a contributing factor when the size of a tumor dramatically decreases.

## Relationship between dN/dS and N across patients

As described above, tumor evolution typically displays a biphasic regime in the relationship between dN/dS and N, consisting of positive correlation in initial phases, apparently, associated with the accumulation of drivers, and negative correlation in late stages, likely, due to the accumulation of deleterious passenger mutations that are eliminated by purifying selection. To further test this conclusion, we examined the relationship between N and dN/dS across the studies. In untreated cancers, dN/dS values across patients were non-monotonic in N, exhibiting an increase at low N and a decrease at high N, in primary and metastatic samples alike (Figs. S1, S2), in agreement with the expectations. By contrast, in the treated cases, this behavior was not, or at least less, apparent (Fig. S3).

The analysis of the multiple myeloma cohort provided a potential explanation for this difference. In this case, we found no clear relationship between dN/dS and N in benign and intermediate stages of the disease, but a non-monotonic behavior in N was evident in disease states (Fig. S9). Furthermore, in samples under continuous treatment, this non-monotonicity was perturbed, presumably, because treatment drives tumor evolution toward the neutral regime (Fig. S9). Thus, the present analysis corroborates on single cohorts the expected non-monotonic behavior of dN/dS in N and additionally suggests that therapy can perturb this behavior as it attracts tumors to the neutral regime. We next examined and validated these conclusions on individual patients.

## Analysis of the relationship between dN/dS and N in individual patients

To further validate the dynamics of dN/dS and demonstrate how it can inform our understanding of the progression of tumors in individual patients and their response to treatment, we analyzed several case studies of patients with multiple samples. We focused on patients with at least 5 samples, where we could evaluate the progression from the treatment-naive state to the metastatic state post-treatment, along with the correlations between dN/dS and N as function of allele frequency (AF) of mutations. Systematic exploration of different AF ranges provides an approximate time signature that increasingly shifts from early, clonal-enriched mutations to late, subclonal-enriched mutations (Fig. 6a). The analysis of these patients provides further evidence of the general non-monotonic character of tumor evolution, whereby signatures of positive correlation between dN/dS and N were mostly observed in early stages, whereas signatures of negative correlation were observed in late stages; in addition, however, patient-specific behaviors were demonstrated (Fig. 6a).

Specifically, we found support of the non-monotonic tumor evolution regime in many patients. For example, such evidence was obtained for two patients with the largest number of samples from the colorectal cancer cohort[42] that included both treated and untreated samples: patient V930, representing the common case of early metastatic dissemination, and patient V750, representing the rarer case of late dissemination, as deduced by the original phylogenetic analysis[42] (Fig. 6b and Fig. S15). In both cases, the dN/dS values increased toward neutrality as the tumor progressed, with the treated metastatic samples being the closest to neutral, corroborating the attraction to the neutral regime in an individual patient. The attraction to the neutral regime as the tumor progressed under treatment blurred to some extent the correlations between dN/dS and N, for the treated samples; however, for the untreated samples, both the positive and the negative correlations, at early and late stages, respectively, were evident and stable (Fig. 6b and Fig. S15).

Although these observations appeared to be robust with respect to the evolutionary path (early vs. late dissemination), the impact of treatment and the timing of driver mutations acquisition varied between cases. For example, in the patient with the largest number of samples from a bladder cancer cohort[52], the effect of chemotherapy on the clonal dynamics was substantial (Fig. 6a). In this patient (P117), there was a significant reduction in N following treatment, in both primary and metastatic lesions, compared to the single treatment-naïve primary sample. However, dN/dS in both the untreated and treated primary samples was close to neutrality, with somewhat lower values in the treated samples (Fig. 6c). After treatment, N increased significantly from the primary samples to the metastatic ones, with a signature of purifying selection that was consistent with the high mutational burden and manifested as lower dN/dS values in the metastatic samples (Fig. 6c). The drastic effect of therapy makes it harder to identify the accumulation of driver mutations and the post-treatment attraction to neutral regime. Nonetheless, the systematic examination of the relationship between dN/dS and N identified a positive correlation between dN/dS and N that was dominant in relatively late stages (Fig. 6a, c). Thus, despite the overall reduced dN/dS values, this dominant positive correlation in these later stages suggests that beyond the advent of the early drivers, accumulation and selection of late (and likely resistant subclonal) drivers continue to occur and shape the evolution of the tumor genome under treatment, consistent with the conclusions of the original study[52].

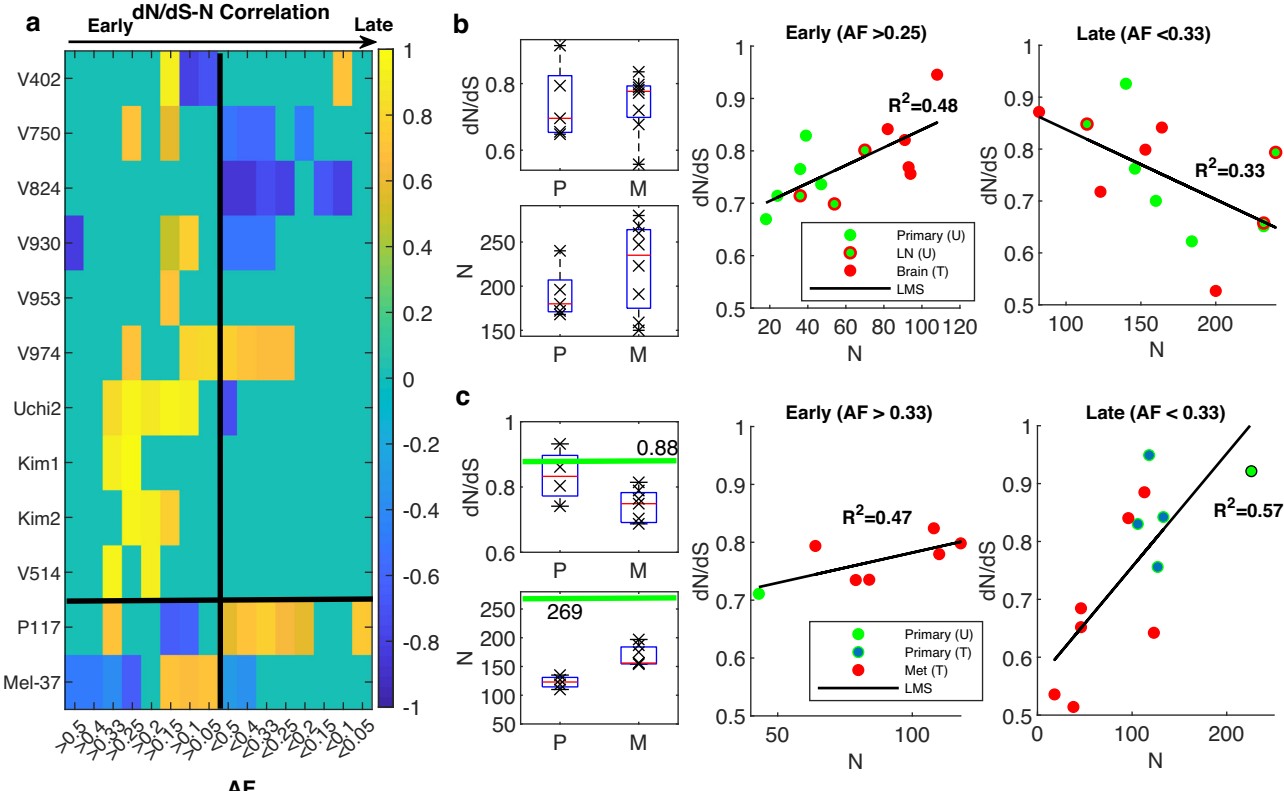

**Fig. 6 | Correlation between *dN/dS* and *N* as function of allele frequency (AF) in single patients. a** Heatmap of Pearson correlation coefficients of the correlation between *dN/dS* and *N* as function of AF, across studies, for patients with at least 5 samples. Black vertical line divides the plane from early (AF>values) to late (AF<values) mutations. The horizontal line separates between studies, with the upper panel corresponding to colorectal patients[42], and below it two single patients: blader patient (P117) with 16 samples from[52] and the single melanoma patient Mel-37 with 37 samples[59]. **b** Example of a colorectal cancer patient (V750, with late metastatic seeding), which includes untreated (U) primary (*n* = 4) and LN (*n* = 3) samples and treated brain metastases (*n* = 5). *dN/dS* and *N* distribution are shown, along with the positive correlation for clonal-enriched early mutations

(AF > 0.25) and the negative correlation for subclonal-enriched late mutations (AF < 0.33). See Fig. S15 for patient V930 (with early metastatic seeding). **c** Example of the bladder patient, exhibiting sharp decline in *N* following therapy with increased load in the metastases, compared to the untreated primary (green lines), as well as dominant positive correlation between *dN/dS* and *N* across most ranges of AF, but specifically in late phases. Only samples that pass the criteria of sufficient no. of mutations in each test are analyzed and shown ("Methods"). Box plots (in **b**, **c**) denote the median (red) and the edges of the box are the 25th and 75th percentiles, with whiskers extending to the most extreme data points. Data points are superimposed (x).

Last, as another demonstration of how the *dN/dS* analysis can help assessing the response to treatment, we analyzed a melanoma patient that was tracked over 9 years, 6 years under immune blockage inhibition, including 37 samples, from the treatment-naïve primary stages, through treatment, to the last metastatic burst[59] (Fig. 6a). The original phylogenetic analysis of the samples identified 6 lineages, from one of which the last metastatic burst originated. Analysis of the data from this patient showed that, despite the general low *dN/dS* values (expected for melanoma due to the typical high mutational burden) and the general low intra-tumor heterogeneity[58] (*cf*. Fig. S6), the lineage that resisted therapy originated from those samples that most closely approached the neutral regime (Fig. S16). Thus, this case study provides additional evidence of how relaxed selection is indicative of tumor progression under treatment, even in cases with a narrow dynamic range of *dN/dS*; furthermore, phases of positive selection in tumor evolution were identified (Fig. 6a and Fig. S16).

## Discussion

The analyses in this study collectively validate that evolution of tumors is non-monotonic, characterized by a bi-phasic regime, whereby at early stages (typically, low *N*), positive selection of drivers dominates, whereas at later stages (typically, high *N*) negative selection acts to eliminate deleterious passengers, with the neutral regime corresponding to the highest tumor fitness (Fig. 7a). At the neutral regime,

the effect of drivers is maximal but the deleterious effect of passengers is not yet pronounced, as predicted by theory[66] and our previous analysis of pan-cancer primary tumors[5,63]. The *dN/dS* value reflects the overall regime of protein level selection across the evolving tumor genome, and as such, determines and shapes the accumulation of more dynamic type of aberrations (e.g., MSI, CNA), often in a compensatory manner[5].

Despite the general attraction to the neutral regime, the strength of protein-level selection (*dN/dS*) substantially varies among individual patients with all tested cancer types, indicating that a tumor can evolve under any of a variety of modes of evolution (purifying selection, positive selection or neutrality). Presumably, the exact regime depends on the balance among various physiological factors including mutational robustness, immune response, microenvironment, somatic evolution in the tissue of origin, and more. A major contribution to the wide range of *dN/dS* values is environmental variation that undoubtedly affects any patient, both through the exposure to different external environments, yielding condition-specific mutational signatures[78,79], and through the internal microenvironment, due to variation in nutrient availability, blood flow, pH, and many other factors[9,80,81]. Under such variable conditions, the efficacy of selection is reduced, so that it becomes difficult for a tumor to maintain specialist traits, blurring the separation between drivers and passengers[82]. Thus, in an ensemble of patients, significant deviations from the dominant

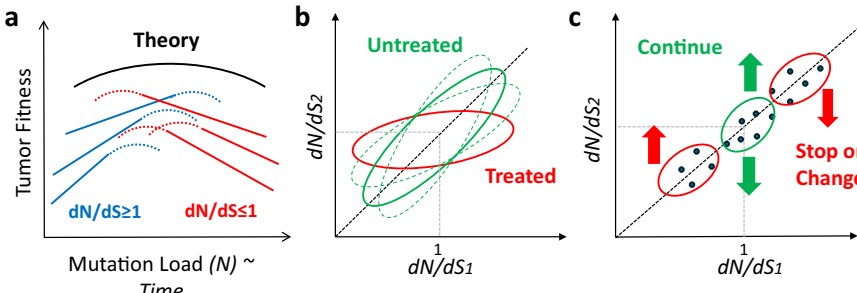

**Fig. 7 | Summary of the findings on tumor evolution regimes and their potential translational value. a** The validated non-monotonic, be-phasic evolutionary regime of tumor evolution, based on theory[66] (black) and empirical evidence across primary cancers[5] (colored lines; each line represent a cancer cohort), showing that tumors operate predominantly near the neutral regime, with signatures of positive selection in early stages (and low $N$) representing the accumulation of drivers, and signatures of negative selection in late stages (and high $N$) indicating the removal of deleterious passengers. **b** The relationship between $dN/dS$ values of early and late samples prior to treatment (green) indicating the roughly linear relationship in the natural progression of primary tumors (solid ellipse), and cancer specific signature in the natural progression to metastases (dashed ellipses). The relationship between the $dN/dS$ values before and after treatment (red), depicting the near universal observed tendency towards neutral evolution (i.e., $dN/dS$ values approach 1) when tumors resist treatment. **c** The translational value of monitoring $dN/dS$ values of biopsies to guide treatment. If a patient starts with $dN/dS$ values around neutrality ($dN/dS \sim 1$), and following treatment, the $dN/dS$ value changes toward a selective regime, the treatment is likely working, and the recommendation would be to continue treatment. In contrast, if $dN/dS$ values before treatment are far from neutrality but steadily approach neutrality following treatment, the treatment is likely to fail, and the recommendation would be to stop or change the treatment.

neutral regime can occur. Strikingly, however, whatever the selection regime of tumor genomes in a patient is, as manifested by the individual-specific $dN/dS$ value, it is highly stable and is effectively linear under natural tumor progression and, to a large extent, under therapy as well (Fig. 7b). Nevertheless, despite this stability, at a finer level, we identified significant dynamic changes in the distribution of $dN/dS$ that provide insights into the natural progression of cancers to metastatic states and especially into the evolution of tumors under treatment, with clinical implications (Fig. 7b). In untreated patients, $dN/dS$ is not affected by natural progression in the primary settings, whereas progression to metastasis can present cancer-specific signatures, such as positive and relaxed selection as identified in the esophageal (cf. Fig. 1b) and breast cohorts (cf. Fig. 3b), respectively, extending previous conclusions that natural progression to metastasis is often underlain by relaxed selection[28]. In contrast, in treated cancers (cf. Figs. 2–4) a nearly universal, significant shift to neutrality was observed, both in hematologic malignancies and in solid tumors (Fig. 7b). In these cases, the trend toward neutrality represents resistance and treatment failure as most (if not all) patients eventually developed resistance to therapy. The post-treatment neutral evolution regime was significantly associated with inferior clinical outcomes, consistent with the validated expectation that neutral evolution corresponds to high tumor fitness. Obviously, the analyzed samples are highly enriched with cases of failed treatment, and no adequate control with successful treatment cases is feasible. Nevertheless, it appears most likely that the shift to neutrality is associated with the emergence of resistance rather than with the effect of the treatment as such. Indeed, analyses of individual patients further elucidated the non-monotonic dynamics of $dN/dS$ and demonstrated how it can inform disease progression and response to treatment, whereby the surviving resistant subclones are likely those that approach the neutral regime.

In summary, we conclude that the universal shift to neutrality in tumor evolution represents an opportunity to harness the evaluation of patients' sample-level selection to guide therapy (Fig. 7c). If a patient's tumor is initially evolving under selection but approaches the neutral regime following treatment, or is stably evolving around neutrality despite therapy, a change in therapy should be considered. Conversely, when a tumor is initially evolving around neutrality, but shifts to a selective regime following treatment, this should be viewed as positive response to therapy. Overall, the results of this work emphasize the importance of quantitatively monitoring the protein-level selection strength ($dN/dS$) in individual patients for understanding tumor evolution and informing therapeutic decisions.

Some questions remain open and worthy of future investigation. For example, different mutational signatures can contribute to the accumulation of mutations and shape $dN/dS$; nonetheless, more specific association between a particular signature and $dN/dS$, and how this might relate to environmental conditions, would require substantially more data of individuals with multiple samples. Possible biases in $dN/dS$ estimates should also be further investigated, in particular, the fitness effect of synonymous mutations that might not be strictly neutral, as well as saturation of synonymous mutations in highly mutated genomes. Such effects could lead to inflated $dN/dS$ values, and taking into account such potential biases would more accurately assess the true strength of selection, especially in highly mutated tumor genomes. Further, in light of the limited number of patients (and samples per patient) that can be clinically tracked under treatment over time, the conclusions of this study should be continuously evaluated as more data become available, to better understand any deviations from the main patterns identified here. Lastly, although the changes in $dN/dS$ values were apparent, the number of genetically distinguishable clones was small and stable (although with substantial dynamic of clone sizes), at least in the multiple myeloma cohort used for validation. This lack of genetic diversity suggests that selection may also act at the epigenetic level, by generating non-genetic diversity, further dividing clones into distinct subpopulations with diverse phenotypic traits and growth strategies that eventually increase (or maintain) the overall tumor fitness. How these types of modifications may occur, under what environmental variable conditions, and what are the relative contributions of the diverse genetic aberrations and non-genetic modifications to tumor fitness and patient prognosis throughout tumor progression, should be considered in the design of future studies, combining genetic and non-genetic measurements.

## Methods

### Datasets

We have selected for analysis all whole-exome sequencing (WES) studies, of either untreated or treated cancer patients, provided that at least two samples per patient existed (e.g., before and after therapy, primary and metastases, or distinct regions) and that for each sample, both the number of non-silent ($N$) and the number of silent ($S$)

mutations were known. These criteria are necessary for estimating the protein-level selection value (*dN/dS*). For initial analysis, these studies included untreated cases of colorectal cancer[42], esophageal cancer[43], and lung cancer[45], and the treated cases of pediatric B-acute lymphoblastic leukemia (ALL)[46], chronic lymphocytic leukemia (CLL)[48], breast[38], bladder[52], and glioblastoma[54], with samples before and after treatment.

For validation, we added to the analysis (**i**) a collection of studies on breast cancer, including untreated cases of early evolution to local LN metastases[29] and to distant metastases[30], as well as treatment resistant cases of ER positive[36] and HER2 positive[37] patients under targeted therapy, with multiple primary samples, and (**ii**) an original cohort of 624 multiple myeloma patients (780 samples) covering all stages, from the benign Monoclonal Gammopathy of Undetermined Significance (MGUS) state and the Smoldering pre-disease state, to the pre-treatment disease state at diagnosis, to the advanced stages of early and late recurrences (relapsed refractory patients with 1–3 and 4 or more lines of therapy, respectively) under continuous treatment (https://doi.org/10.7303/syn53268838). Patients were consented to the TCC Protocol, the Moffitt Cancer Center's institutional biorepository (MCC#14690; Advarra IRB Pro00014441). Sex (and race and ethnicity) was not considered in the study design and was based on all available patients. Self-reporting from electronic health records, was provided to us by an honest broker, with balanced demographic of 56% male and 44% female. Patients agreed to donate additional bone marrow aspirate during a clinical bone marrow biopsy procedure, blood samples, and grant access to their medical records. These samples were enriched for CD138+ cells (MM, tumor) and subjected to research use only (RUO) grade whole-exome sequencing under the ORIEN/AVATAR program. Investigators obtained signed informed consent from all patients who were enrolled in the protocol MCC14690 conducted at the H. Lee Moffitt Cancer Center & Research Institute, as approved by the institutional review board. To this end, patient samples were used in accordance with the Declaration of Helsinki, International Ethical Guidelines for Biomedical Research Involving Human Subjects (CIOMS), Belmont Report, and U.S. Common Rule. The medical records were de-identified, where serum/urine electrophoresis data was analyzed to estimate clinical response.

### Whole exome sequencing and mutational calling of original cohort of multiple myeloma samples

**Sample preparation.** Fresh BM aspirate cells were enriched for CD138 expression using Miltenyi (Bergisch Gladbach, Germany) 130-051-301 antibody-conjugated magnetic beads. $1.0 \times 106$ viably frozen CD138$^+$ cells were shipped for molecular analysis in the context of the ORIEN/AVATAR program.

**Nucleic acid extraction.** For frozen tissue DNA extraction, Qiagen QIASymphony DNA purification was performed, generating 213 bp average insert size.

**DNA whole exome sequencing.** Preparation of Whole Exome Sequencing (WES) libraries involved hybrid capture using an enhanced IDT WES and Nimblegen SeqCap EZ kits (38.7 Mb) with additional custom designed probes for double coverage of 440 cancer genes. Library hybridization was performed at either single or 8-plex and sequenced on an Illumina NovaSeq 6000 instrument, generating 100 bp paired reads. WES was performed on tumor/normal matched samples with the normal covered at 100X and the tumor covered at 300X (additional 440 cancer genes covered at 600X) depth. Both tumor/normal concordance and gender identity QC checks were performed. Minimum threshold for hybrid selection is >80% of bases with >20X fold coverage; ORIEN/AVATAR WES libraries typically meet or exceed 90% of bases with >50X fold coverage for tumor and 90% of bases with >30X fold coverage for normal samples.

**Phylogenetic trees and mutational signatures.** To generate phylogenetic trees for patients with multiple biopsies, we used the package MesKit for R (version 1.1.2), using as input mutational data in maf format, and Neighbor-Joining as the method for tree construction. Mutational signatures for individual branches of phylogenetic trees were generated using MesKit package for the signature set COSMIC V2. For the mutational signatures for entire cohort, we have used R package MafTools (version 2.20.0) to identify the contribution of COSMIC V2 mutational signatures for each sample, and these values where averaged for all samples within each disease state.

**Copy number alterations analysis.** To assess the ploidy level in each sample, we used sequenza-utils (version 3.0.0) to generate seqz files from tumor and germline bam files. The R package sequenza (version 3.0.0) was used to generate segments with differential copy numbers and cntools (version 1.30.0) was used to determine the copy numbers of individual genes.

**Microsatellite instability (MSI) analysis.** We used the R package MSISensor2 (version 0.1; https://github.com/niu-lab/msisensor2) to calculate a microsatellite instability score. Tumor genomes with score above 3.5 were considered unstable (i.e., MSI), and genomes with score below it were considered stable (i.e., MSS).

**Clonality FISH plots for sequential samples.** To assess the no. of clones in each sample, we used the R packages sciClone (version 1.1.1) and fishPlot (version 0.5.2), to infer and depict clonal composition in sequential biopsies.

### Evaluating the evolutionary state of tumor genomes by *N* and *dN/dS*

**Background.** Tumor evolution can be depicted as an evolving tree of lineages, some of which would die due to deleterious mutations, and some would grow due to mutations conferring selective advantage[2]. The two main parameters that control this process are the number of non-silent (*N*) mutations and the strength of selection, which can be estimated at the molecular level by the *dN/dS* ratio[83]. This is because *N* naturally determines the phylogenetic relationship between the branches in the tree (e.g., shared and private mutations), and selection determines the population dynamics (i.e., which subclones would die or grow) and thereby the shape of the tree. Further, *dN/dS* is directly related to the scaled (by population size) selection coefficient of a mutation[84]. And, in cancer, regardless of the mode of evolution (e.g., linear, branched, punctuated or neutral)[85,86], over sufficiently long time scales, *N* can be viewed as a proxy for the time since the birth of the neoplastic cell because *N* is relatively monotonic in time, except for fluctuations due to selective sweeps[87]. Thus, given the time of progression, *N* can be related to the mutation rate[5,63]. These properties render *N* and *dN/dS* principal parameters that control not only the fate of cancer cell populations, but also generally the fate of any organism, whereby negative selection (*dN/dS* < 1) predominantly indicates how large populations (e.g., prokaryotes) adapt by eliminating deleterious mutations (and respective subpopulations), while neutral evolution (*dN/dS* ≈ 1) and positive selection (*dN/dS* > 1) predominantly indicates the paramount role of random genetic drift in small populations (e.g., vertebrates) and the expansion of neutral or advantages mutations (and respective subpopulations)[5,12,76,77,88,89].

**Rationale.** The information on *N* is most efficiently extracted by phylogenetic reconstruction of tumor evolution, as demonstrated by studies we analyze here. Therefore, here we sought to focus on the analysis of *dN/dS* and test if it contained signatures of the effects of tumor progression and or therapy. Two main approaches have been developed to estimate *dN/dS*, either at the gene level[67] or at the genome level[63]. Both methods require sufficient statistics of the number of

$N$ and $S$ mutations, for accurate estimation of $dN/dS$. At the gene level, this is achieved by integrating mutations across patients. This assumes that gene identity and function are the main determinants of selection, such that inter-patient heterogeneity can be averaged out. At the genome level (e.g., patient or sample level), sufficient statistics can be attained by integrating the number of mutations in the genome. This assumes that the specific physiology and (micro)environment of each patient (or sample) are the main determinants of selection, considering their paramount roles in tumor evolution[9,78], such that integration can be performed across genes. Thus, here, we evaluated $dN/dS$ at the genome level, also because, as a per-sample variable, it allows for (**i**) coping with small number of patients (and samples per patient) in a typical study analyzed here, and importantly (**ii**), for correlating $dN/dS$ with patients' survival clinical outcomes, because information across patients (that could lead to overfit) is not used.

**Formula.** To estimate $dN/dS$, the method treats the genome as one concatenated sequence and calculates the number of non-synonymous ($nN$) and synonymous ($nS$) sites across the genome, normalizing the number of $N$ and $S$ mutations across the genome, such that $dN/dS = N/nN/S/nS$[63]. In using this formula, we assume that synonymous mutations are neutral and are not saturated (i.e., a site is not mutated more than once). This procedure can be applied to estimate $dN/dS$ for a sufficiently large set of genes (typically >100), provided there exist enough mutations for the estimation. For example, in driver genes, one often finds elevated $dN/dS$ ratios[63]. This is demonstrated here on the large cohort of multiple myeloma (Fig S17). Nonetheless, heavily mutated genes, including driver genes, may exhibit low $dN/dS$ ratios, reflecting negative selection of deleterious passengers at high $N$ (cf. Fig. 7a). Similarly, the same metric is used to estimate $dN/dS$ along a phylogenetic tree, where shared mutations (in the trunk of a tree) typically represent clonal mutations, and private mutations (at the branches of a tree) are typically enriched with sub-clonal mutations. Due to the extreme sparseness of mutations in the genomes, to increase statistical power of $dN/dS$ estimation in a region, background silent mutations (defined as those in genes harboring only $S$ mutations), are attributed proportionally to the size of the region for which $dN/dS$ is evaluated. For more information see ref. 63.

## Main analysis procedures

**Linear regression fits and tests.** $N$ and $dN/dS$ were evaluated based on the WES mutation data of, and as provided by, the analyzed studies. We compare $dN/dS$ (and $N$) between two samples in a patient, $dN/dS\_1$ vs. $dN/dS\_2$, where $dN/dS\_1$ represents the earlier sample, such that depending on the dataset, the comparison is of samples before and after diagnosis, before and after treatment, primary vs. metastasis, or primary vs. primary. When the temporal order of samples is not known (e.g., primary vs. primary), we assume that an early sample contains less mutations than a more evolved sample. To test the significance of the difference between two linear regressions models we used one-way ANOVA test and Fisher statistics. To test the significance of the deviation of a single linear regression from a strict linear relationship, we generated an artificial linear reference model, that represents the variance in the data, using noise in the response that is equal to the residuals of the least mean square linear regression of the tested fit, and applied ANOVA F-test.

**Errors and biases in the estimation of dN/dS.** To avoid biases in the estimation of $dN/dS$ due to sampling errors in a patient (or a sample), we excluded from analysis cases with very small numbers of mutations. A threshold of 10 mutations was derived by matching the TCGA database mutational data to a neutral model, the dominant regime in tumor evolution[63,67,68,70] (Fig. S18) and identifying the minimal number of mutations below which there was a significant deviation from

expectation (Fig. S19). Cases with fewer mutations than this threshold were excluded from the analysis. We estimated the error in the evaluation of $dN/dS$ to be of the order 0.1, using the cohort of Multiple Myeloma, where several patients had multiple samples (of the same status) taken at the same time and sequenced separately (Fig. S20). The low error on the estimates of $dN/dS$ is also reflected in the general invariance of the wide distributions of $dN/dS$ across patients in each cohort, since large errors would necessarily blur the stability of the $dN/dS$ values across samples.

**Assessing the robustness of dN/dS to various factors.** to ensure that our results are not significantly biased due to potential potential confounding factors, we tested the stability of the $dN/dS$ estimates with respect to CNA, MSI and tumor purity, in the large original validating cohort of multiple myeloma patients. We have previously shown that the $dN/dS$ metric is insensitive to CNA, whereby the distribution of $dN/dS$ in diploid regions is statistically indistinguishable from $dN/dS$ distribution in regions affected by CNA[63]. Nonetheless, to confirm that this was indeed the case, here, we estimated $dN/dS$ for different levels of CNA. We found that $dN/dS$ distribution in genes with high CNA (75th percentile) was not statistically significant different from $dN/dS$ distribution in genes with low CNA (Fig. S21). Similarly, the distributions of $dN/dS$ in diploid regions and regions affected by CNA were highly similar and statistically indistinguishable (Fig. S21), consistent with our previous analysis. Further, a similar analysis was conducted with respect to the extent of MSI, finding that $dN/dS$ distribution was also highly stable with respect to the classification of tumor genomes into MSI and MSS (Fig. S21). Last, although sequencing was performed after selection and enrichment of cancer plasma cells, extremely low tumor purity prior to enrichment could potentially lead to biases in the estimation of $dN/dS$. Nonetheless, we found that the $dN/dS$ estimates were highly stable and were not significantly affected by the level of tumor purity (Fig. S22).

**Survival analysis.** To correlate variables with clinical data we applied the Kaplan-Meier analysis[90] to the censored survival data of patients and used log-rank test[91] to estimate the $P$-value of difference between two survival curves. For paired samples, before and after treatment, to assess the association of neutral evolution with patients' survival, we classified patients to 'neutral', defined as those in the neutral regime or approaching it post treatment, and to 'escape', defined as those far-from or escaping the neutral regime, using two measures: (i) DIS: the distance from neutrality post treatment, given by the absolute value of $1-dN/dS\_2$ (ii) DELTA ($\Delta$): the amount of change in dN/dS post treatment, given by dN/dS\_2 – dN/dS\_1. Using these measures we systematically explored all possible regimes, whereby patients were classified as 'escape' if they exhibit high DIS, or high DELTA (provided $dN/dS\_2 < 0.9$ or $dN/dS\_2 > 1.1$). Thus, high DELTA represents unstable cases, whereby $dN/dS$ can change dramatically (e.g., from negative selection to positive selection) and end up outside the strict neutral regime ($1 \pm 0.1$).

**Effect of therapies.** To estimate the effect of different therapies, we conducted a Cox Proportional Hazard Analysis, whereby each possible drug was represented as an independent binary variable (1 if present in the regimen before the biopsy, and 0 otherwise). The outcome variable was the distance of $dN/dS$ values of each sample from neutrality (DIS). We used the function coxph in R survival package (version 3.7-0) to assess the contribution of each drug to the deviation from neutrality, as denoted by the Hazard Ratio (HR).

## Reporting summary

Further information on research design is available in the Nature Portfolio Reporting Summary linked to this article.

## Data availability

Source files analyzed in this study are publicly available from the supplementary material of the respective studies, which include untreated colorectal cancer[42] (https://www.nature.com/articles/s41588-019-0423-x#Sec22) esophageal cancer[43] (https://www.nature.com/articles/s41467-019-09255-1#Sec36) and lung cancer[45] (https://www.nature.com/articles/s41467-021-25787-x#Sec24), and treated ALL[46] (https://www.nature.com/articles/ncomms7604#Sec22), CLL[48] (https://www.nature.com/articles/s41467-018-03170-7#Sec25), breast cancer[38] (https://www.nature.com/articles/s41588-018-0287-5#Sec40), bladder cancer[52] (https://www.nature.com/articles/ng.3692#Sec26) and glioblastoma[54] (https://www.nature.com/articles/ng.3590#Sec29). The additional studies on breast cancer include natural progression to local LN[29] (https://aacrjournals.org/clincancerres/article/24/19/4763/80918/The-Spatiotemporal-Evolution-of-Lymph-Node-Spread) and to distant metastases[30] (https://aacrjournals.org/clincancerres/article/23/15/4402/257274/Genetic-Heterogeneity-in-Therapy-Naive-Synchronous), as well as treated cases of ER positive[36] (https://www.nature.com/articles/ncomms12498#Sec25) and HER2 positive[37] (https://www.nature.com/articles/s41467-019-08593-4#Sec17), under targeted therapy. Source mutational and clinical files analyzed in this study and the code that analyzes these files and reproduces the main results and figures, are also available in Supplementary Software 1. For the multiple myeloma original cohort, WES data and the processed MAF file for multiple myeloma samples collected at H. Lee Moffitt Cancer Center & Research Institute with annotated mutations can be downloaded from the following folder in Synapse, https://doi.org/10.7303/syn53268838. The processed data is also available in Supplementary Software 1. The raw data of this cohort is not publicly available, as it was generated through private funding by Aster Insights (www.asterinsights.com) in collaboration with the Oncology Research Information Exchange Network (ORIEN, www.oriencancer.org). Access to the raw data (FASTQ/BAM) of this study may be granted upon request, which should be submitted to the corresponding author (A.S.S) and at https://researchdatarequest.orienavatar.com/. Raw data must be used for non-profit research purposes in accordance with Moffitt's Total Cancer Care protocol 14690. The time frame to gain access to the raw data can typically take up to a few months, following review of the user's research statement, and will be made temporarily available for download.

## Code availability

The data analyzed in this study and the analysis scripts that fully reproduce the main results and figures are provided as Supplementary Software 1 (CODE.zip) under exclusive copyright.

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

## Acknowledgements

We sincerely thank our multiple myeloma patients and their families for donating their samples for research purposes, and members of the Multiple Myeloma Working Group at Moffitt Cancer Center. We also thank members of the Koonin group at the NLM, as well as Aljandro Schaffer, Joo Sang Lee, Sridhar Hannenhalli and Eytan Ruppin from the NCI for feedback and comments. We are also grateful for the assistance and service of the Tissue and Cancer Pharmacodynamics-Pharmacokinetics cores of Moffitt Cancer Center, and Cores of Aster Insights (formerly known as M2Gen). This work was supported by personal donations from the Pentecost Family Myeloma Research Center (to A.S.S.) and in parts by the H. Lee Moffitt Cancer Center Physical Sciences in Oncology (PSOC) Grant 1U54CA193489-01A1 (to A.S.S.), H. Lee Moffitt Cancer Center's Team Science Grant (A.S.S.), Miles for Moffitt Foundation (A.S.S.), and 2021 Multiple Myeloma Research Foundation (MMRF) Research Fellowship Award (P.R.S.). This research was made possible through the Oncology Research Information Exchange Network (ORIEN)/Avatar Project in collaboration with Aster Insights, the Total Cancer Care protocol at the H. Lee Moffitt Cancer Center & Research Institute; an NCI designated Comprehensive Cancer Center (P30-CA076292). We gratefully acknowledge funding from Physical Sciences Oncology Network at the National Cancer Institute (grant U01CA261841). E.P., Y.I.W. and E.V.K. are supported by intramural funds from the US Department of Health and Human Services (to the National Library of Medicine).

## Author contributions

E.P., Y.I.W. and E.V.K. conceived and designed the study. E.P., P.R.S. and A.S.S. performed the analysis. E.P., P.R.S., Y.I.W., R.R.C., M.D., K.H.S, A.S.S. and E.V.K. analyzed data. P.R.S., R.R.C., K.H.S and A.S.S. contributed new reagents/analytic tools. E.P., A.S.S. and E.V.K. wrote the manuscript that was edited and approved by all authors.

## Funding

## Competing interests

The authors declare no competing interests.
