## [Peer Review File · Nature Communications]

Genome-level Selection in Tumors as a Universal Marker of Resistance to Therapy

Corresponding Author: Dr Erez Persi

Version 0:

Reviewer comments:

Reviewer #1

(Remarks to the Author)

The authors study cancer evolution and evidence of positive, neutral or negative selection before and after treatment and its predictive value on treatment outcomes. They make use of several previously published diverse cancer datasets and give an account of “N” (number of non-synonymous mutations) and “dN/dS” (the rate of non-synonymous mutations to the rate of synonymous mutations) in tumors. They suggest that before treatment, dN/dS which could be indicative of positive (>1), negative (<1) or neutral selection (~ 1) is invariant irrespective of timepoints (early or late) or location (primary or metastatic) across cancers; however, a regime change post-treatment towards neutral evolution from previously positive or negative selection signals treatment failure and faster time to relapse or worse prognosis.

The work presented is interesting and novel. Here are some points to consider:

Major Points:

1. How does the dN/dS values in this study compare to dN/dS values published by other studies? For instance, Figure 1A seems to show the majority of the samples under negative selection in colorectal cancer whereas (PMID: 36289336) report evidence of mostly positive selection. What could be the reason for this difference?
Could the authors provide more details in the Methods section on how dN/dS was calculated?
2. The assumption of a linear correlation with a slope of 1 between the dN/dS ratios of tumors from primary and metastatic sites in untreated cancers may not be universally applicable, as demonstrated by the breast cancer case in Figure 3B. This observation raises concerns about the generalizability of the authors' claim that pre-treatment tumor genomes are uniformly stable and invariant to tumor progression. To strengthen this conclusion, it would be necessary to analyze a larger and an even more diverse cohort of cancers. This broader analysis could better categorize tumors into those that are mostly stable versus those that exhibit significant genomic instability prior to treatment.
3. Please provide coefficients of independent variables along with the p-values within each of the correlation plots. This will help assess the impact of potential outliers and test the robustness of the correlation. For instance in Figure 2, with CLL it seems to be a single point (C812) that is driving the drift towards neutrality post-treatment. Can you also provide risk tables at the bottom of Kaplan meier curves for Figure 2 so that the numbers at risk are exposed?
4. Could the observed dN/dS values in the post-treatment samples be influenced by contamination with normal cells or reduced tumor purity, potentially driving these values toward neutrality? Would be helpful to discuss any steps taken to account for this in their analysis.
5. Since in most cancers chosen for pre-treatment versus post-treatment dN/dS analysis, most of the pre-treatment samples have values already close to neutrality, it is a little hard to see the shift towards neutrality post-treatment (Fig2). The authors themselves acknowledge this issue in the main text. To strengthen the study's conclusions, have the authors considered performing power calculations to determine the appropriate sample size needed to detect the hypothesized shift with sufficient statistical power, particularly given the small effect size observed? Additionally, it would be informative to perform a more granular analysis by examining the dN/dS ratios on a more stratified basis. Specifically, could the authors first conduct a paired analysis pre-treatment and post-treatment samples that were under negative selection at pre-treatment, showing the the Kaplan-Meier curves of those who shifted towards neutrality versus those who maintained negative selection and/or changed regime to positively selected, assessing the observed trend, and then conduct a similar analysis for those that were under positive selection at pre-treatment? This stratified approach could provide deeper insights into the dynamics of selection pressure in response to treatment.
6. Could the authors provide a comparison of their model, which suggests a shift toward neutrality post-treatment as an

indicator of worse outcomes, against established biomarkers such as tumor burden, high-risk mutations, and other clinical risk factors? An assessment of the predictive performance of their model relative to these existing biomarkers would strengthen the validity of their approach. This is particularly important given the borderline significance observed in the Kaplan-Meier curve presented in Figure 4B ($p = 0.055$). How does the predictive value of this shift toward neutrality compare in terms of sensitivity, specificity, and overall prognostic utility?

7. Can the authors comment on the observed clonal diversity and dN/dS relation that they observe before and after treatment? Is the shift to a neutral regime associated with lower clonal diversity where positive selection was previously observed and vice-versa?

Minor:

1. Kindly provide tables with the x, y variables that were used to make the scatter plots in csv or excel format for reproducibility of plots and statistics. Currently the code provided is in matlab file format and I could not find the data tables for all cases.
2. "ALL" missing in the title of plot 2A.
3. Figure 4C, x-axis ticks missing axis "time in days?"

Reviewer #2

(Remarks to the Author)

Genome-level Selection in Tumors as a Universal Marker of Resistance to Therapy

The authors of the study titled "Genome-level Selection in Tumors as a Universal Marker of Resistance to Therapy" have demonstrated that neutral evolution occurs in various cancer types after treatment failure, and they associate this with a poorer prognosis. This study included both primary and metastatic cases on the same individuals, pre and post treatment samples from multiple tumor types. Finally, they propose a strategy in which global dN/dS measurements could serve as a tool to monitor patient response to treatment.

Major Comments:

1. The manuscript presents an intriguing perspective on the evolutionary biology of tumors. The idea that all point mutations in a population of tumor cells could be used to predict their fitness stage at any given time point is compelling. However, while this argument touches on a fundamental aspect of tumor evolution, focusing exclusively on point mutations (SNVs and indels) is a significant limitation. Natural selection affects various types of genomic alterations, and therefore, a comprehensive framework of selection acting upon multiple layers of heritable alterations is needed. Here, the study focuses on the dN/dS ratio, but how might other genomic alterations, such as copy number variations (CNVs) and inversions, behave under these conditions?
2. Why would an increase in genetic diversity in certain cancers under chemotherapy suggest a more neutral dN/dS ratio? Does the emergence of subclones post-treatment indicate a stronger selective advantage for a particular clone, followed by a phase where only neutral events accumulate, thus resulting in a more neutral dN/dS ratio when using all mutations? These questions naturally lead to the need for clarity on how tumor heterogeneity is measured—whether it is based on subclonal diversity or total genetic variation.
3. Similarly, when estimating dN/dS values using mutations from the same patient but in different regions or at different time points, it would be insightful to compare the proportion of shared mutations with those that are distinct. If enough mutations are available, calculating dN/dS for shared versus private mutations could reveal shifts in selection patterns across different spatial and temporal contexts. This can be done by establishing a dual dN/dS system—one for all mutations and another for all minus private mutations. This could clarify patterns of directional selection more clearly. Moreover, all analyses should ideally consider mutations based on VAF-based methods to identify clonality, clonal diversity, or intratumor heterogeneity (ITH). Is the shift towards neutrality associated with a reduction in subclones due to large clonal expansion?
4. It is also essential to ensure that deviations from neutrality in some cohorts are not due to different strategies for somatic variant calling. It is known that germline contamination can push dN/dS values below 1.
5. The manuscript suggests that tumors often exhibit neutral evolution and increased fitness following treatment failure. However, how do these tumors show higher fitness if their dN/dS values do not indicate clonal expansions or positive selection? Perhaps analyzing dN/dS ratios with and without known driver genes for each tumor type could reveal underlying patterns of fitness advantage. Also, for which treatments this would be the case? It would be valuable to discuss whether these findings could be applied to other datasets and compare the effect of different treatments.
6. On a technical note, N (mutation burden or TMB) should be closely associated with the confidence intervals obtained when calculating dN/dS. The absence of these intervals makes the manuscript's interpretations difficult to fully accept, even if they are accurate. Including some measure of uncertainty based on the total number of mutations would strengthen the analysis. It would also be useful to compare TMB, N, and dN/dS across all cases.
7. The timing of mutation acquisition is also an important factor when considering tumor evolution. A model/figure demonstrating the relationship between the variables of time and the dN/dS obtained would benefit the reader in understanding how these affect the results of early versus late. Specifically, it is important to determine a priori whether mutations are early or late in all the analyses using an orthogonal method.
8. Another critical point is mutational signatures. The authors note that in the validation set, metastases also show signs of relaxed selection, particularly when no strong mutational signature is present. It would be beneficial to estimate dN/dS specific to different mutational signatures (a simple analysis could compare dN/dS for clock-like versus non-clock-like signatures).
9. One of the major weaknesses of the manuscript is the lack of appropriate controls to test the hypothesis of increased neutrality associated to higher fitness for tumors post treatment. This limitation unfortunately makes the interpretation of selection measurements less applicable to clinical settings. If the authors could identify appropriate datasets to demonstrate a response, or perhaps test their findings with known organoid models treated with various drugs, the results would be more

robust.

10. For clarity to the broad audience of Nature Communications, the hypothesis currently presented in Figure 6 should be moved to Figure 1 and explained with a simple model of tumor evolution. This model should include potential reasons behind changes in the selective regime and its interaction with treatment and evolution.

Minor Comments:

- In the sentence: “decrease in dN/dS ($dN/dS < 1$), particularly in recurring stages (Figure 4A), presumably a consequence of the accumulation of deleterious passenger mutations,” it is important to clarify that it is not the accumulation of deleterious mutations that drives $dN/dS < 1$, but rather the removal of these mutations from the population. This distinction should be clarified throughout the text.
- Line 477: The manuscript states that dN/dS requires sufficient numbers of N and S mutations for accurate estimations. Are there any tumor types where this metric may not perform well, such as those with low TMB? Discussing this limitation could provide a clearer understanding of the method's applicability.
- Line 218: Correct the typo “poatients” to “patients.”
- Provide an explanation for the observed differences between primary vs. metastasis and late_primary vs. early_primary samples in Supplementary Figure 1. Why do esophageal tumors appear more genetically similar compared to colorectal samples?
- Supplementary Figure 5: Correct the typo in the title of the right panel from “Relpase” to “Relapse.”
- Supplementary Figure 12: Increase the panel sizes for better clarity. Additionally, ensure consistency in the capitalization of group labels (e.g., “untreated” should be consistently lowercase or uppercase).
- Supplementary Figure 17: Increase the size of the graphs to better distinguish the observed vs. expected mutations across various tumor types.
- Line 126-129: This section refers to one patient, but Figure 1A refers to multiple patients. Please briefly clarify the comparison.

Reviewer #3

(Remarks to the Author)

Reviewer #4

(Remarks to the Author)

In their manuscript entitled “Genome-level Selection in Tumors as a Universal Marker of Resistance to Therapy”, Persi and colleagues investigate global dN/dS values in treated and untreated cancers. They observe a tendency for treated cancers to have lower dN/dS values in a variety of settings.

I was intrigued by the abstract of the manuscript, which contained some very surprising statements, such as “we observe a nearly universal shift toward neutral evolution following treatment failure”. The reason I found this surprising and intriguing is that it is not at all obvious why tumors would “switch” to neutral evolution after treatment failure, what exactly that even means biologically, or how it could be effectively quantified. The authors end up not really answering these questions and ultimately, I don't think the data presented in the paper support their conclusions. Although the statistics in the manuscript left me with many unanswered questions, I believe overall that dN/dS values in treated tumors drop as the authors claim. However, I think the simplest explanation for this phenomenon – which is also in line with the larger literature – is that treatment causes cancers to go through bottlenecks, from which they emerge with more (mostly neutral) mutations that have accumulated during tumor growth. These mutations depress the dN/dS ratio because regular chemotherapy treatment typically does not elicit resistance through specific driver/resistance mutations, and even targeted treatment resistance is usually mediated by a very small set of functional mutations. However, the treatment bottlenecks expose many neutral variants that have accumulated during years and decades of growth. I think that is essentially what the authors are seeing here.

It is universally acknowledged that most selection happens on the way from normal tissue to the tumor founder cell, i.e. most driver mutations that would elevate dN/dS are happening along the tumor trunk. After that, this driver contingent stays relatively invariant, even during metastasis (see eg Reiter et al. Science 2018) and many people believe that many tumors essentially evolve neutrally after the birth of the founder cell (see e.g. Williams et al, Nature Genetics 2016). Evidence of subclonal evolution is hard to demonstrate, it typically requires multi-region sequencing at sufficient depth. So I am not sure what the authors really mean when they say there is a “shift to neutrality” after treatment failure – such a thing cannot be proven by a crude measure like genome-wide dN/dS in my opinion. It would require careful multi-region sampling in treated and untreated tumors to demonstrate that in untreated tumors, evidence of subclonal evolution beyond the variants that are obviously enriched in the tumor trunk is happening, and it would require that the authors show that the same is not true after treatment – a tall order indeed. In the absence of such an analysis, I think that the claims this manuscript makes are not substantiated.

A few more specific comments:

The regression approach that yields p-values throughout the paper is highly unusual and cannot be understood in detail from the methods. It also yields rather suspect results, like a p-value of 0 for the regression line in figure 2B where only a single data point shows a drop in dN/dS while all other data points are clustered together near 1,1. Can the authors replicate

their statistics with more standard methods, e.g. a Wilcoxon signed-rank test would work in this scenario?

In general, I find the paper's statistical arguments are weak in places and sometimes not well justified. For example, the text claims "The post-treatment dN/dS values close to neutrality were associated with the worst outcome (rapid relapse), suggesting that the shift to neutrality is associated with higher tumor fitness and treatment failure (Fig. S4)." – but Fig. S4 has no statistical analysis, and the claim of a quadratic relationship between time to relapse and dN/dS is not believable, as it's essentially driven by a single data point. The authors also do not explain why a quadratic relationship would be expected based on any biological considerations.

Reviewer #5

(Remarks to the Author)

Version 1:

Reviewer comments:

Reviewer #1

(Remarks to the Author)

The authors have satisfactorily answered my questions and meaningfully revised the manuscript. I have no further suggestions or comments

(Remarks on code availability)

Reviewer #2

(Remarks to the Author)

The authors have answered my concerns

(Remarks on code availability)

Ok

Reviewer #3

(Remarks to the Author)

(Remarks on code availability)

Reviewer #4

(Remarks to the Author)

I regret to say that this rebuttal and revision did not significantly change my assessment of the manuscript, primarily because I did not feel that the authors engaged in a good faith effort to answer the raised points. Most importantly, they failed to address our primary concern that dN/dS may be confounded by mutation burden. As we explained, we expect that post-treatment samples have a substantially greater (passenger) mutation burden than pre-treatment samples, raising the concern that signals of selection are 'drowned out' by non-functional mutations to a greater degree post-treatment. The authors did not perform analyses to alleviate this central concern, nor did they directly respond to it. Instead, they turned to an analysis of tumor size which is of questionable relevance given the uncertain correlation between tumor size and mutation burden. I note that no direct replies or analyses were produced in the rebuttal, and the revised manuscript did not highlight revisions, it is therefore largely unclear which changes – if any – the authors may have made in response to the review. Similarly, the concern that a lot of the results in the paper were driven by single or extremely limited data points (a real issue in Figures 2 and 3) was dismissed by the authors.

(Remarks on code availability)

Reviewer #5

(Remarks to the Author)

(Remarks on code availability)

Version 2:

Reviewer comments:

Reviewer #2

(Remarks to the Author)

I think the authors have satisfactorily addressed all comments

(Remarks on code availability)

The code seems to provide all the relevant material for reproduction including a README.

Reviewer's Comments:

Reviewer #1 (Remarks to the Author): Expert in cancer genomics, multiple myeloma genomics and clinical research

The authors study cancer evolution and evidence of positive, neutral or negative selection before and after treatment and its predictive value on treatment outcomes. They make use of several previously published diverse cancer datasets and give an account of “N” (number of non-synonymous mutations) and “dN/dS” (the rate of non-synonymous mutations to the rate of synonymous mutations) in tumors. They suggest that before treatment, dN/dS which could be indicative of positive (>1), negative (<1) or neutral selection (~1) is invariant irrespective of timepoints (early or late) or location (primary or metastatic) across cancers; however, a regime change post-treatment towards neutral evolution from previously positive or negative selection signals treatment failure and faster time to relapse or worse prognosis. The work presented is interesting and novel. Here are some points to consider:

We thank the reviewer for the constructive comments and suggestions, which we have found important and insightful. In the revised manuscript, we have addressed all of them, as explained below, point by point.

Major Points:

1. How does the dN/dS values in this study compare to dN/dS values published by other studies? For instance, Figure 1A seems to show the majority of the samples under negative selection in colorectal cancer whereas (PMID: 36289336) report evidence of mostly **positive selection**. What could be the reason for this difference? Could the authors provide more details in the Methods section on how dN/dS was calculated?

The metric of dN/dS at the genome level that we use here was previously studied by us on the pan cancer TCGA data of primary tumors and was compared to other studies (Persi et al, PNAS 2018). The reason for the discrepancy pointed out above is the difference between estimates of dN/dS at the gene level and dN/dS at the genome level, as explained and discussed in **Methods**. In the study referred to above (i.e., PMID: 36289336), positive selection is inferred for a specific set of driver genes, where positive selection is indeed expected. This is expected also by our metric when applied to a set of driver genes, as we have shown previously (Persi et al, 2018).

In the revised manuscript, we extended the **Methods** section considerably to provide more details on the metric of dN/dS, how it is calculated and included analysis of dN/dS in numerous contexts, in particular, clonal vs. subclonal mutations, early vs. late mutations, as well as driver vs. other genes. In **Methods** we focus on the technical accuracy and stability of dN/dS as a metric, demonstrated against various factors. In the **Results**, the ways in which the dynamics of dN/dS elucidates tumor evolution in these contexts are demonstrated throughout; in particular see revised **Figures 5 and 6**, which validate expectations of when positive and negative signatures are expected (explained in revised **Figure 7A**).

2. The assumption of a linear correlation with a slope of 1 between the dN/dS ratios of tumors from primary and metastatic sites in untreated cancers may not be universally applicable, as demonstrated by the breast cancer case in Figure 3B. This observation raises concerns about the generalizability of the authors' claim that pre-treatment tumor genomes are uniformly stable and invariant to tumor progression. To strengthen this conclusion, it would be necessary to analyze a larger and an even more diverse cohort of cancers. This broader analysis could better **categorize tumors into** those that are **mostly stable** versus those that exhibit **significant genomic instability** prior to treatment.

We acknowledged and emphasized (from the Abstract to the Discussion) that in untreated cancers there exist cancer-specific signatures in the natural progression to metastases (compared with regional primary progression), such as positive selection in esophageal (**Figure 1B**), and indeed, relaxed selection in breast cancer (**Figure 3B**); in breast cancer, this occurs mostly in advanced cases, while early stages are more linear. The invariance (or close to linear) property refers to untreated cancers in primary settings, as demonstrated throughout the paper on colorectal, esophageal, lung, breast, melanoma, and multiple myeloma. This is now better described in the revised manuscript, including in abstract and summary (**Figure 7B**). We also note that these results are compatible with those of previous pan-cancer studies (Hu et al, 2020) (**Discussion**). The novelty here is rather the universal shift to neutrality under treatment (with slope much lower than 1, typically < 0.6 as in **Figure 2**), while in untreated cancers the slopes “fluctuate” around 1 (typically ± 0.3 , considering cancer-specific deviations). The differences between the slopes are now clearer in all revised **Figures 1-4**.

We should also mention that to validate the linear regime in untreated cases vs. the shift to neutrality in treated cases, it was more important to analyze untreated vs. treated cases in the same cancer type (rather than include more case studies) – which we have done for the case of Multiple myeloma (**Figure 4**). The overall analyses cover 10 cancers (colorectal, esophageal, lung, breast, ALL, CLL, bladder, glioblastoma, melanoma and multiple myeloma), whereby we have included all studies we could identify that include both N and S mutations across the whole genome (necessary for the estimation of dN/dS at the sample level; many relevant studies (which we refer to) do not include this information and this was the sole criterion for study selection (**Methods**)).

The case of unstable genomes, such as microsatellite instability (MSI) or high copy number alteration (CNA) is generally interesting. However, it is not expected to affect our analysis. **First**, diverse cancers have different levels of instability, yet the linear regime in untreated cancers (slopes 1 ± 0.3) and the shift to neutrality in treated cancers (slopes < 0.6) appear unaffected by these differences. For example, colorectal cancer typically exhibits high MSI, in contrast to lung cancer with typical stable genomes (low MSI) (see Hause et al, Nat med 2016) – yet, the slope is close to 1 in both cases (**Figure 1**). **Second**, technically, the metric of dN/dS that we use is insensitive to CNA. This was evident in our previous analysis (Persi et al, PNAS, 2018), where we found only small differences in dN/dS in diploid regions vs. regions affected by CNA. **Third**, dN/dS based on point mutations reflects the general selective regime of the tumor genome, and

accordingly, it determines and shapes the accumulation of larger and more dynamic aberrations, often in a compensatory fashion (see our Perspective in Nat Rev Gen, 2021). For example, MSI and CNA accumulate under (weak) positive and neutral regimes (if/when there are not enough drivers), while chromosomal instability accumulates at later stages under negative selection (to mitigate the effect of deleterious passengers).

In the revised manuscript, we now refer to some of these aspects in the revised **Discussion**. Further, we have included additional analyses, where we measure both the level of MSI and CNA in the large and validating cohort of multiple myeloma. This analysis confirms that dN/dS metric is largely unaffected by the genome stability status, whereby the distributions of dN/dS in high MSI vs. low MSI and in high CNA vs. low CNA are statistically indistinguishable (see **Methods** and **Fig. S21**). The analysis also demonstrates the accumulation of MSI and CNA in the sequential tumor progression of MM, under the effectively dominant neutral regime (see **Figure 4A**).

See also reply to related **point 1 by Review 2**.

3. Please provide coefficients of independent variables along with the p-values within each of the correlation plots. This will help assess the impact of potential outliers and test the robustness of the correlation. For instance, in Figure 2, with CLL it seems to be a single point (C812) that is driving the drift towards neutrality post-treatment. Can you also **provide risk tables** at the bottom of **Kaplan meier curves** for Figure 2 so that the numbers at risk are exposed?

In the revised manuscript, we have included in all Figures the linear regression coefficients with their p-values, as well as confidence interval curves. As mentioned above (in point 2), the differences between untreated (with slopes around 1 +/- ~ 0.3) and treated cases (with slopes < 0.6) are now more evident. Regarding CLL, the shift comes from several points beyond C812, including C811 and those cases that 'escape' neutrality (associated with better prognosis); nonetheless, given the small number of patients (in a study), we have analyzed several studies, and performed two levels of validation (with published and original data).

We now have added the number of cases for each KM curve in each survival analysis. The risk table (for each curve) is simply the multiplication of this number by the values of survival in the y-axis; we did not include it for clarity of the figures/space availability. Further, the Code.zip package allows for full reproducibility including the complete statistics of the linear regressions, ANOVA tests and KM survival analysis (beyond what is provided in the figures).

4. Could the observed dN/dS values in the post-treatment samples be **influenced by contamination with normal cells or reduced tumor purity**, potentially driving these values toward neutrality? Would be helpful to discuss any steps taken to account for this in their analysis.

Tumor purity should not affect the dN/dS estimates. This is because cancer cells are purified from the biopsies, and only then sent for sequencing. Thus, the mutation calling (after passing all quality control requirements) is based on purified cancer cells and should not lead to any biases.

Nonetheless, generally, it is possible that samples with very high contamination (low tumor purity) pre-enrichment may be subjected to higher noise/inaccuracy in the mutational calling, and thereby affect dN/dS estimation although not necessarily in the form of a bias toward neutrality. For example, we would expect the opposite in MM because high tumor purity in the marrow indicates tumor growth, which we link here to neutrality.

In the revised manuscript, as an additional test, we included estimates of tumor purity in the large Multiple-Myeloma cohort (used for validation) and checked for possible biases in dN/dS. The results show that there are no biases in the distribution of dN/dS even when samples with very low tumor purity (<10% plasma cells, prior to enrichment with CD-138) are compared with the rest of the sample. The results of these tests are now provided as part of the extended **Methods** and **Fig. S22**.

5. Since in most cancers chosen for pre-treatment versus post-treatment dN/dS analysis, most of the pre-treatment samples have values already close to neutrality, it is a little hard to see the shift towards neutrality post-treatment (Fig2). The authors themselves acknowledge this issue in the main text. To strengthen the study's conclusions, have the authors considered **performing power calculations** to determine the appropriate sample size needed to detect the hypothesized shift with sufficient statistical power, particularly given the small effect size observed? Additionally, it would be informative to perform a more **granular analysis by examining the dN/dS ratios on a more stratified basis**. Specifically, could the authors **first conduct a paired analysis** pre-treatment and post-treatment samples that were under negative selection at pre-treatment, **showing the Kaplan-Meier curves of those who shifted towards neutrality versus** those who maintained negative selection and/or changed regime to positively selected, assessing the observed trend, and then conduct a similar analysis for those that were under positive selection at pre-treatment? This stratified approach could **provide deeper insights into the dynamics of selection pressure in response to treatment**.

The shift to neutrality is manifested by a change in dN/dS in opposing direction (i.e., an increase for values starting below 1 and a decrease for values starting above 1), leading to a 'rotation' of the regression fit. The statistical significance of this rotation is now evaluated by two measures: **(i)** how the fit deviates from a strict linear fit (without rotation), as denoted by the F-statistics of ANOVA test (see **Methods**, and P-values in main text) and **(ii)** by slopes of the regressions being significantly smaller than 1 (typically < 0.6) and their P-values (following point 2 above). Both measures indicate that the shift is significant. Our analysis is based on existing data from the respective studies (including the original MM cohort), thus, we did not calculate the sample size beforehand but rather use all existing data to maximize the statistical power and significance of results.

We appreciate the important suggestion to include additional granular analysis of dN/dS with respect to patients' survival. This is relevant for the 2 large cohorts with sufficient survival data: glioblastoma in **Figure 2** and MM in **Figure 4**. **In the revised manuscript**, we have explored more thoroughly how the classification of patients based on dN/dS associates with clinical outcomes for these 2 cohorts. Specifically, we now explore systematically:

(i) for a cohort, we examined all possible groups of samples by selecting those in the range $dN/dS = \mu \pm \sigma$ and compare them with all other patients (whereby $\mu \approx 1$ denotes those close to neutrality). The new results show clearly that the association of neutrality with worse prognosis is more robust than we previously reported. For example, in the MM it is now evident that there is a clear shift to neutrality post treatment and that this shift is associated with worse prognosis, which represents a hill in the tumor fitness landscape (**Figure 4C**), while clusters of patients that deviate from neutrality represent valleys in the tumor fitness landscape. Note that the minimal P-values are much lower than what we initially reported by selecting only one (reasonable) point (of 1 ± 0.2).

(ii) for paired samples (before-after treatment), we classify patients into those that are steadily neutral or become neutral post treatment and those that are 'far from' or 'escape' neutrality using two measures (see **Methods**): distance from neutrality post-treatment (DIS), given by: $\text{abs}(1 - dNdS_{\text{post}})$ and the change in dN/dS (DELTA) given by: $dNdS_{\text{post}} - dNdS_{\text{pre}}$ reflecting how (un)stable is the positive/negative selection regime in each patient. Using these measures, patients exhibiting large DIS and DELTA are classified as 'far from' or escaping neutrality, which are expected to have better prognosis. The results show that this is indeed the case (see **Figure 2E** and **Figure 4E**). Note that this analysis reflects more accurately the suggested paired analysis (partitioning to quarters is not feasible as there are too few data points for a meaningful survival analysis).

Overall, we believe these new analyses now better demonstrate how the metric of dN/dS can be used as a powerful tool to explore each case study, for better calibration and potential use in the clinics.

6. Could the authors provide a comparison of their model, which suggests a shift toward neutrality post-treatment as an indicator of worse outcomes, against established biomarkers such as tumor burden, high-risk mutations, and other clinical risk factors? An assessment of the predictive performance of their model relative to these existing biomarkers would strengthen the validity of their approach. This is particularly important given the borderline significance observed in the Kaplan-Meier curve presented in Figure 4B ($p = 0.055$). How does the predictive value of this shift toward neutrality compare in terms of sensitivity, specificity, and overall prognostic utility?

We addressed in our analysis, when possible, how dN/dS is associated with other clinical data in the respective studies (e.g., time to relapse, tumor size, response to treatment) – these analyses support the association of neutrality with short time to relapse (Figs. S4-S5), tumor growth (Fig. S14) and resistance to treatment (**Figures 2E** and **4D**). The improved survival analysis in the **revised manuscript**, as explained in the reply to point 5 above (and revised new **Figure 4**), shows that the association of neutrality with worse prognosis is robust and highly significant (with much lower P-values than reported initially).

We believe that the relationship to tumor burden should be understood in the context of the underlying framework, namely, that at low N (early stages / clonal / trunk), dN/dS increases with

N (as a signature of positive selection of drivers) whereas at high N (late stages / subclonal / branches), dN/dS decreases with N (as a signature of negative selection of deleterious passengers), with neutral evolution (intermediate N, time) being associated with the highest tumor fitness. **In the revised manuscript**, these features are now validated in **Figure 5** (based on phylogenetic trees of individuals with many samples) and **Figure 6** (based on the relationship of dN/dS with N, as a function of VAF). The validated framework is now also described in the summary (**Figure 7A**), beyond the descriptions in **Introduction** and **Methods**.

Obviously, our analyses suggest that dN/dS is an additional biomarker (and not the only one) that correlates with tumor fitness and clinical outcome. In the clinic, it should be considered together with other relevant biomarkers, such as ISS for MM, which are different from the tumor molecular characteristics we discuss here (for example, ISS in MM is based on blood measures and not the bone marrow) and thus may or may not be associated with dN/dS and its dynamics.

7. Can the authors comment on the observed **clonal diversity and dN/dS** relation that they observe before and after treatment? Is the shift to a neutral regime associated with lower clonal diversity where positive selection was previously observed and vice-versa?

In the revised manuscript, we extend and include additional analyses to address the association of dN/dS with clonality and clonal dynamics:

(i) based on phylogenetic tree reconstruction of patients with many samples, where we estimate dN/dS in clonal/trunk mutations and in subclonal/branch mutations (see **Figure 5**). This analysis shows that dN/dS is much greater than 1, in a clear sign of positive selection for clonal mutations, while mutations at the tree branches are subject to negative selection, such that the overall regime of all mutations is close to neutrality. This holds true universally for treated and untreated cancers and regardless of the trees' shapes (e.g., short or long branches). This analysis is demonstrated in revised **Figure 5** (for MM under treatment) and **Fig. S13** (for untreated colorectal and esophageal cancers). It suggests that, due to the strong negative selection at the branches, one would expect reduced clonality. However, this does not necessarily translate into a reduced number of clones, as explained next.

(ii) based on FISH plots (applied for the MM cohort), which show the number of clones in the sequential progression of MM tumors (see **Figure 5**). This analysis shows that the number of clones is much less dynamic than dN/dS, but that dynamics nevertheless shows interesting features of changes in clone sizes. Thus, it is hard to associate dN/dS (and neutrality) with clonal reduction (i.e., no. of clones), even if this is the case in some patients. Furthermore, these findings suggest that selection of specific clones may also occur beyond the genetic level, that is, epigenetically. We discuss that this type of analysis remains relevant and should be the subject of future studies (see **Discussion**).

(iii) based on the relationship between dN/dS and N as a function of VAF, whereby mutations with high VAF correspond to early (clonal-rich) mutations and those with low VAF correspond to late (subclonal-rich) mutations (see **Figure 6**). This analysis is done for cohorts with the highest

number of patients in the relevant studies (colorectal, Bladder, Melanoma) such that the correlation between dN/dS and N can be estimated. This analysis demonstrates signature of positive selection in early stages (positive correlation between dN/dS and N) and signatures of negative selection at late stages (negative correlation between dN/dS and N). Nonetheless, the analysis also demonstrates that these features are case-specific and how they can be exposed by such analysis. For example, in a bladder patient (P117 in **Figure 6**), the accumulation of mutations post-treatment at low AF indicates selection of resistant mutations post treatment (consistent with the conclusion of the original study). In a melanoma patient (Mel-37 in **Figure 6**), the dynamics of dN/dS shows dominance of negative selection across time (with typically low values, due to the high burden); yet, a direct comparison with the phylogenetic tree and clonal compositions of the samples of this patient indicate that the increase in dN/dS (towards neutrality) occurs in the clone that eventually bursted into multiple metastases leading to death (see **Fig. S16**).

Overall, these analyses not only validate the underlying framework (now shown in **Figure 7A**), but also demonstrate that dN/dS metric is a useful, reliable and accurate tool to explore clonal dynamics and identify when positive selection of drivers occurs.

Minor:

1. Kindly provide tables with the x, y variables that were used to make the scatter plots in csv or excel format for reproducibility of plots and statistics. Currently the code provided is in matlab file format and I could not find the data tables for all cases.

For the convenience of the reviewer and potential readers, we now provide with the CODE.zip package also an xlsx table with the requested values (see file dNdS_summary.xlsx).

2. "ALL" missing in the title of plot 2A.

Corrected.

3. Figure 4C, x-axis ticks missing axis "time in days?"

Corrected.

Reviewer #2 (Remarks to the Author): Expert in cancer genomics and evolution, mathematical modelling, and response to therapy

The authors of the study titled "Genome-level Selection in Tumors as a Universal Marker of Resistance to Therapy" have demonstrated that neutral evolution occurs in various cancer types after treatment failure, and they associate this with a poorer prognosis. This study included both primary and metastatic cases on the same individuals, pre and post treatment samples from multiple tumor types. Finally, they propose a strategy in which global dN/dS measurements could serve as a tool to monitor patient response to treatment.

We thank the reviewer for the careful review and constructive suggestions, which we have found important and insightful. In the revised manuscript, we have addressed all of them, as explained below, point by point.

Major Comments:

1. The manuscript presents an intriguing perspective on the evolutionary biology of tumors. The idea that all point mutations in a population of tumor cells could be used to predict their fitness stage at any given time point is compelling. However, while this argument touches on a fundamental aspect of tumor evolution, focusing exclusively on point mutations (SNVs and indels) is a significant limitation. **Natural selection affects various types of genomic alterations**, and therefore, a comprehensive framework of selection acting upon multiple layers of heritable alterations is needed. Here, the study focuses on the dN/dS ratio, but how might other genomic alterations, such as **copy number variations (CNVs) and inversions**, behave under these conditions?

First, dN/dS is a measure that can be applied only to point mutations, and indeed, it captures a fundamental aspect – it assesses the strength of the protein-level selection acting on a genome. As such, dN/dS also largely shapes how other types of (larger and more dynamic) aberrations (e.g., MSI, CNA and CIN) are accumulated in the tumor genome in different phases, as we have demonstrated previously, often in a compensatory manner (Persi et al, 2021). Under neutrality, many aberrations would be significant (e.g., MSI, CNA), while in later phases, when the selection regime may turn to purifying selection (due to burden of deleterious passengers, and/or activation of immune response), larger aberration such as CIN can accumulate as means to mitigate the effects of high burden (Lopez et al, Nat gen 2020, Alfieri et al, Nat com 2023). Negative selection in turn reduces the level of small aberrations (Mlecnik et al, Immunity 2016). Thus, dN/dS appears fundamental also in shaping other aberrations.

Second, there's considerable diversity of aberrations across different cancers and individuals, such as typical high-MSI in colorectal cancer but low-MSI in lung cancer (Hause et al, Nat med 2016), yet, the stability of dN/dS in untreated cases and the shift towards neutral evolution under treatment appear nearly universal. This is likely a consequence of the fundamental feature (captured by dN/dS) whereby tumor fitness is high under neutral evolution, such that tumors are 'attracted' to this regime (see also Persi et al, 2018 which corroborates empirically theoretical prediction by McFarland et al 2013).

In the revised manuscript, we now include complete characterization of MSI and CNA in the large MM sample used for validation, verifying that the values of dN/dS in high MSI or high CNA cases are not statistically distinguishable from those with stable genome. This is consistent with (and in addition to) our previous analysis on the TCGA database, covering 23 cancers (Persi et al, PNAS, 2018). Thus, our analysis (and the metric of dN/dS) is technically robust to these aspects. The fundamental relationship between dN/dS and how it shapes other aberrations is now briefly described in the **Discussion**.

See also related **point 2** by **Reviewer 1**

2. Why would **an increase in genetic diversity** in certain cancers under chemotherapy suggest a more **neutral dN/dS ratio**? Does the emergence of **subclones post-treatment** indicate a stronger **selective advantage for a particular clone**, followed by a phase where only neutral events accumulate, thus resulting in a more neutral dN/dS ratio when using all mutations? These questions naturally lead to the need for **clarity on how tumor heterogeneity is measured**—whether it is based on subclonal diversity or total genetic variation.

We consider all the mutations in the genome for the general assessment of dN/dS, while of course it is important to address the dynamics of dN/dS with respect to the phase of evolution and clonality. This is done in the analysis of single patients with multiple samples, which we have revised and extended.

In the revised manuscript, we now include 3 levels of analysis of the relationship between dN/dS and clonality (see details in **point 7 by Reviewer 1**). Briefly, (i) via the phylogenetic trees, estimating dN/dS in clonal/trunk/shared mutations vs. dN/dS in subclonal/branch/private mutations, across studies (ii) via analysis of clonal (FISH) plots of the MM cohort used for validation, along the phylogenetic tree, and (iii) via the relationship between dN/dS and N as function of AF (accounting for early vs. late mutations), across studies. The results are presented in revised **Figures 5 and 6** and related, complementary **Figs. S13, S15 and S16**.

The analyses show that generally, trunk/shared mutations are subject to positive selection (presumably, a sign of accumulation of drivers), while branch/private mutations are subjected to purifying selection (presumably a sign of accumulation of passengers negatively affecting tumor fitness), such that overall, considering all mutations, the regime is close to neutral (**Figure 5**). Nonetheless, there are case-specific features, whereby in a bladder cancer patient, we identify positive selection of post-treatment mutations (via the positive correlation of dN/dS with N, for sub-clonal mutations with low AF), in contrast to the colorectal cancer patients where positive selection is identified for mutations with high AF (representing the trunk), and in a melanoma patient, negative selection is dominant at every phase (due to extremely high N), yet the clone that eventually bursted into multiple metastases exhibits relaxed selection (**Figure 6**).

Overall, these results, combined with the survival analysis (cf. **Figure 4 and 2**), validate our working assumption (now described in **Figure 7A**; based on theory, corroborated in previous analysis) that tumor fitness is highest in the neutral regime (where there is maximum drive while the effect of deleterious passenger mutations is not yet strong). This appears to be the principal reason for the general “attraction” to the neutral regime, which becomes stronger post treatment, while deviations from this general trend could occur in individual patients.

3. Similarly, when estimating dN/dS values using mutations **from the same patient but in different regions or at different time points**, it would be insightful to compare the proportion of shared mutations with those that are distinct. **If enough mutations are available, calculating dN/dS for shared versus private mutations could reveal shifts in selection patterns** across different spatial and temporal contexts. This can be done by establishing a dual dN/dS system—one for all mutations and another for all minus private mutations. This could clarify patterns of directional selection more clearly. Moreover, **all analyses should ideally consider mutations based on VAF-based methods to identify clonality**, clonal diversity, or intratumor heterogeneity (ITH). Is the shift towards neutrality associated with a reduction in subclones due to large clonal expansion?

The combined extended analysis described in the reply to **point 2** above (and **point 7** of **Reviewer 1**) answers these important and insightful questions. In particular, the analysis of shared vs. private mutations reveals dominant positive selection of shared mutations, while

negative selection is dominant in private mutations, resulting in an overall (close to) neutral regime (see **Figure 5**, and **Fig. S13**). This feature is reproduced in the analysis of dN/dS-N correlation as function of VAF (see **Figure 6**).

The results show that the shift to neutrality indicates strong negative selection that counteracts the positive selection in earlier phases, which would indeed suggest reduced clonality; However, this does not necessarily translate into reduction in the number of clones which appears less dynamic (see **Figure 5**). The clonal dynamic is interesting and will continue to be the subject of future work. It may suggest the impact/importance of epigenetic selection of phenotypic traits – we added these points to the **Discussion**.

4. It is also essential to ensure that deviations from neutrality in some cohorts are not due to different strategies for **somatic variant calling**. It is known that germline contamination can push dN/dS values below 1.

In this work, we analyze published data and an original dataset of MM. In the published data sets, virtually all sequencing is based on comparing samples with a paired normal cell (usually blood) and various controls are applied to ensure accurate mutational calling (the basis for dN/dS estimation).

In the revised manuscript, we also added measures of tumor purity (% of plasma cells) in the analysis of the large MM cohort used for validation. The results show that dN/dS values are not affected even in extreme cases of very low tumor purity because enrichment of cancer cells is performed prior to sequencing (see **Methods** and **Fig SI22**, as well as reply to related **point 4** by **Reviewer 1**).

Further, it should be stressed that we assess errors in the estimation of dN/dS and avoid biases using additional measures (see reply to **point 6** below). These aspects are now better highlighted in the extended, revised **Methods**.

5. The manuscript suggests that tumors often exhibit neutral evolution and increased fitness following treatment failure. However, how do these tumors show higher fitness if their dN/dS values do not indicate clonal expansions or positive selection? Perhaps **analyzing dN/dS ratios with and without known driver genes** for each tumor type could reveal underlying patterns of fitness advantage. Also, **for which treatments this would be the case?** It would be valuable to discuss whether these findings could be applied to other datasets and **compare the effect of different treatments**.

We believe that the neutral regime (over the entire genome) corresponds to the highest tumor fitness because it reflects a balance between the beneficial effects of drivers and possible deleterious effects of passengers (see Persi et al, 2018 and Persi et al, 2021) which is consistent with theoretical expectations (see McFarland et al, 2013) – this hypothesis is now emphasized in **Figure 7A**. The selection regime may indeed vary with clonal dynamics (as demonstrated on single patients), and it can differ among genes as well. Indeed, by applying our metric to driver genes, signatures of positive selection can be revealed, as we have shown in our previous

analysis of the TCGA data set (Persi et al, 2018) – see also reply to **point 1** of **Reviewer 1**. Nonetheless, note that drivers are sometimes identified by the high frequency of mutations in them, and this may lead to signatures of negative selection (because high N implies the existence of deleterious passengers that would lead to negative selection as a mechanism for their elimination from the population) according to our working assumption (presented in **Figure 7A**). **In the revised manuscript**, we now discuss and include estimation of dN/dS in driver genes for the MM cohort showing these features (see **Methods** and **Fig. S17**).

Regarding the effect of different treatments, it is possible to detect positive or negative selection regimes via the relationship between dN/dS and N, as we demonstrate on single patients – this analysis also shows that the selection of drivers is case-specific, either before treatment (e.g., colorectal) or post-treatment (e.g., a specific bladder patient) in **Figure 6**. Further, we analyze the response to treatment, when possible. Nonetheless, the comment regarding the effect of different treatments is interesting and we aim to address this as well in future work. **In the revised manuscript**, we analyzed the effects of different treatments/drugs on dN/dS in the large MM cohort. The analysis shows that most of the drugs pushed tumors closer to the neutral regime (and resistance), while only a few drove them away from the neutral regime (expected to be more beneficial) – these results are shown in revised **Figure 4**.

6. On a technical note, **N (mutation burden or TMB) should be closely associated with the confidence intervals** obtained when calculating dN/dS. The absence of these intervals makes the manuscript's interpretations difficult to fully accept, even if they are accurate. Including some measure of uncertainty based on the total number of mutations would strengthen the analysis. It would also be useful to compare TMB, N, and dN/dS across all cases.

In **Methods**, we explain how we account for errors in dN/dS and avoid possible biases using various measures. **First**, when the number of mutations is too low, the confidence in the estimate of dN/dS is indeed reduced, due to sampling bias, and this leads to a known bias towards lower values of dN/dS. We avoid such biases by removing cases with too small numbers of mutations – this is based on analysis of the TCGA data, matching it to a neutral model, and deducing the threshold below which a bias may occur. Obviously, the accuracy in dN/dS improves with the number of mutations, such that above this threshold, no significant biases are expected. **Second**, we estimate the error on dN/dS empirically, by sending the same biopsies to sequencing in parallel in the MM cohort – we show that this error is small (of the order of 0.1) compared to the much larger changes that occur naturally, with time and with disease progression. The respective analysis and results are provided in **Methods** and **Figs. S18-S20**.

We dedicated two sections in **Results** to demonstrate and discuss the relationship between N-dN/dS: for the analysis of all cohorts and in the extended analysis, we also demonstrate this relationship in single patients as a function of VAF (revised **Figure 6**). The analyses demonstrate that the underlying hypothesis is valid – with dN/dS increasing at low N (dominance of drivers) and decreasing at high N (dominance of passengers). Yet, there exist case-specific features, such as post-treatment positive selection of drivers, as in the case of a bladder cancer patient P117 (detected via the positive correlation of dN/dS with N). We conclude that generally, treatment blurs the relationship between dN/dS and N, consistently with the 'attraction' to the neutral regime.

7. The timing of mutation acquisition is also an important factor when considering tumor evolution. A model/figure demonstrating the relationship between the variables of time and the dN/dS obtained would benefit the reader in understanding how these affect the results of **early versus late**. Specifically, it is important to **determine a priori whether mutations are early or late** in all the analyses **using an orthogonal method**.

In the revised manuscript, we now demonstrate systematically in the extended analysis of individual patients with many samples (**Figure 6**) the differences in dN/dS for early (high VAF) vs. late (low VAF) mutations. The results largely validate the assumption of positive selection of drivers in early stages (dN/dS increase with N for low AF) and negative selection at later stages (to counteract the accumulation of passengers), but also reveal case-specific features as mentioned above in point 6. This is consistent with the additional analysis in the revised manuscript that address the relationship between dN/dS and clonality (described above in the reply to **point 2 and 3**, as well as **point 7** by **Reviewer 1**).

8. Another critical point is mutational signatures. The authors note that in the validation set, metastases also show signs of relaxed selection, particularly when no strong mutational signature is present. It would be beneficial to estimate dN/dS specific to different mutational signatures (a simple analysis could compare dN/dS for clock-like versus non-clock-like signatures).

We agree that the study of mutational signatures is generally interesting and important. However, the mutational signatures represent an additional layer of partitioning the mutation data, such that it cannot be readily combined with the dN/dS metric. Specifically, the signatures are estimated from the aggregate mutation data (partitioned to nearest nucleotides context, i.e. 96 bins), so that it is impossible to assign a specific single mutation to a signature (a single mutation can belong to many signatures). Thus, we cannot select a list of mutations assigned to a signature to estimate dN/dS specifically for that signature. Nonetheless, it is possible to estimate the principal signatures (and dN/dS) along the branches of phylogenetic trees of individuals with many samples, which we do in the revised manuscript.

In the revised manuscript, we include analysis of the dominant signatures in the large cohort of MM patients used for validation, as well as in the different branches of the phylogenetic trees for individuals with multiple samples, along with the analysis of shared vs. private mutations (see **Figure 5**). The results reveal that in this cohort, beyond the expected signatures (COSMIC 1 and 2) that occur in all cancers, the dominant signatures are related to DNA damage (COSMIC 3, 6 and 15) and tobacco (COSMIC 4 and 29). Beyond that, we point out that across studies, signatures vary considerably, yet the effects we discuss (in particular, the shift to neutrality) appear quite universal. Nonetheless, we believe this issue should be further investigated in future studies, with more datasets of patients with many samples each, to allow for deeper insights – as now mentioned in the revised **Discussion**.

9. One of the major weaknesses of the manuscript is the lack of appropriate controls to test the hypothesis of increased neutrality associated to higher fitness for tumors post treatment. This limitation unfortunately makes the **interpretation of selection measurements** less applicable to clinical settings. If the authors could identify **appropriate datasets to demonstrate a response**,

or perhaps test their findings with known organoid models **treated with various drugs**, the results would be more robust.

We associate the selection regime with tumor fitness based, primarily on the correlation of dN/dS with patient survival, which relies on theory and empirical evidence (**Figure 7A**). This is unequivocally demonstrated in the extended analysis (see reply to **point 5** by **Reviewer 1**) both on published data (i.e., Glioblastoma – **Figure 2**) as well as on original data (i.e., MM – **Figure 4**). In both cases, the shift to a neutral regime is associated with worse prognosis (while escaping/being far from it is associated with better prognosis). An ideal control should include healthy tissues in the same cancer type - and this is done in the validating MM cohort, which tracks patients from the healthy (benign) stages to the advanced disease stages (recurrence). In this cohort, we demonstrate a close to linear relationship in healthy untreated tissues (MGUS and Smoldering), while in active disease stages, dN/dS shifts to neutrality under treatment, concomitantly with worse prognosis and resistance to treatment. In the revised manuscript, we now also include analysis of dN/dS changes with respect to different treatments/drugs, indicating that most drugs push dN/dS closer to neutrality (see revised **Figure 4**). The use of organoid models, however, is beyond the scope of this study. Moreover, the survival analyses are consistent with additional analyses of clinical data (in various datasets)– showing that the shift to neutrality post treatment is associated with tumor growth, resistance to treatment and faster relapse (see for examples, **Figs. S4, S5, S10, S14** and **S16**).

The interpretation of dN/dS dynamics and its applicability are simple and straightforward, as explained in **Figure 7C**. Nonetheless, the revised extended analysis emphasizes how this metric can and should be used as an investigative tool for each cohort (**Figure 4**) and individual (**Figures 5-6**), to detect cancer- and patient- specific behaviors.

10. For clarity to the broad audience of Nature Communications, the hypothesis currently presented in Figure 6 should be moved to Figure 1 and **explained with a simple model of tumor evolution**. This model should include potential reasons behind changes in the selective regime and its interaction with treatment and evolution.

The working assumption is now added as **Figure 7A**. It is based on theoretical results (McFarland et al, 2013, 2014) corroborated empirically in our previous work (Persi et al, 2018 and 2021) by analysis of TCGA data along with clinical outcomes of patients – indicating that tumor fitness is highest around neutrality. Because this model is also explained in the **Introduction**, we believe that it is optimal to recapitulate it in the summary **Figure 7**, so that the main results of the study (in **Figure 7B** and **7C**) can be more easily understood along with the model, and because the entire analysis further validates this model (in particular, the clonality analysis in **Figures 5** and **6** and the survival analysis in **Figures 2** and **4**).

Minor Comments:

- In the sentence: “decrease in dN/dS ($dN/dS < 1$), particularly in recurring stages (Figure 4A), presumably a consequence of the accumulation of deleterious passenger mutations,” it is important to clarify that it is not the accumulation of deleterious mutations that drives $dN/dS < 1$,

but rather the removal of these mutations from the population. This distinction should be clarified throughout the text.

Corrected

- Line 477: The manuscript states that dN/dS requires sufficient numbers of N and S mutations for accurate estimations. Are there any tumor types where this metric may not perform well, such as those with low TMB? Discussing this limitation could provide a clearer understanding of the method's applicability.

The threshold below which biases occur is assessed systematically by matching the entire TCGA database to a neutral model, and cases with too low numbers of mutations are removed from the analysis, as explained in **Methods** (and related SI) - see also our reply to **point 6** above.

- Line 218: Correct the typo “poatients” to “patients.”

Corrected.

- Provide an explanation for the observed differences between primary vs. metastasis and late_primary vs. early_primary samples in Supplementary Figure 1. Why do esophageal tumors appear more genetically similar compared to colorectal samples?

The existence of cancer specific signatures, often in the form of relaxed (or positive) selection, in the natural progression to metastasis (vs. within primary tumors) is emphasized throughout the paper (see reply to **point 2** by **Reviewer 1**). In the primary setting, one expects a roughly linear regime which is the case in all the tested cancers including esophageal, lung, breast, MM, and melanoma. In the case of colorectal –2 points indeed lead to a deviation from a strict linear fit (slope of 0.67), however, according to the ANOVA test, this fit is not statistically distinguishable from the more linear P-M progression (slope of 0.8), that is, in colorectal, dN/dS in both P-P and P-M progression is roughly linear.

- Supplementary Figure 5: Correct the typo in the title of the right panel from “Relpase” to “Relapse.”

Corrected.

- Supplementary Figure 12: Increase the panel sizes for better clarity. Additionally, ensure consistency in the capitalization of group labels (e.g., “untreated” should be consistently lowercase or uppercase).

We ensure the resolution is sufficiently high – one can zoom in for clarity. Unless we miss something, as far as we can see, all titles are consistently uppercase in this figure (now S14).

- Supplementary Figure 17: Increase the size of the graphs to better distinguish the observed vs. expected mutations across various tumor types.

As for Figure S12 (now S14), it is hard to increase the size of the subplots in this quite loaded figure. However, the resolution is sufficiently high for one to be able to zoom in and see the differences.

- Line 126-129: This section refers to one patient, but Figure 1A refers to multiple patients. Please briefly clarify the comparison.

We are not sure what this comment is about. The section refers to the colorectal cohort.

Reviewer #3 (Remarks to the Author)

We appreciate this initiative, as well as Nature acknowledging it.

Reviewer #4 (Remarks to the Author): Expert in cancer genomics and evolution, mathematical modelling, and translational research

In their manuscript entitled “Genome-level Selection in Tumors as a Universal Marker of Resistance to Therapy”, Persi and colleagues investigate global dN/dS values in treated and untreated cancers. They observe a tendency for treated cancers to have lower dN/dS values in a variety of settings.

We thank the reviewer for the careful review of our manuscript and the important, insightful comments. However, our main observation is somewhat different: we observe a shift to neutrality post treatment, meaning either an increase in dN/dS (from values below 1) or a decrease in dN/dS (from values above 1), not a general tendency to lower values.

I was intrigued by the abstract of the manuscript, which contained some very surprising statements, such as “we observe a nearly universal shift toward neutral evolution following treatment failure”. The reason I found this surprising and intriguing is that it is not at all obvious why tumors would “switch” to neutral evolution after treatment failure, what exactly that even means biologically, or how it could be effectively quantified. **The authors end up not really answering these questions and ultimately, I don't think the data presented in the paper support their conclusions.**

The shift to neutrality is manifested by slopes of regression fits being typically below 0.6 in treated and resistant cancers. In contrast, natural progression in untreated cancers exhibits much higher slopes, typically of 1 ± 0.3 ; the significant deviations are attributed to the progression to metastasis with cancer-specific signatures. This is demonstrated in **Figure 1** (untreated) and **2** (treated) across cancers, and then validated in **Figure 3** and **4** within the same cancer type.

The association of the post treatment shift to neutrality with tumor fitness is unequivocally demonstrated in the revised manuscript, by the extended survival analysis (see reply to **point 5** by **Reviewer 1**) showing the shift exists and correlates with worse prognosis (see **Figure 2E** and **Figure 4B-C**). This is further supported by analysis of a variety of clinical data across studies, including tumor growth, response to treatment and time to relapse (**Figs. S4, S5, S10, S14, S16**).

The biological reason for this is that under neutral evolution the fitness of tumors is highest, such that tumors tend to thrive in, and be attracted to, this regime. This is because in this regime there is maximal accumulation of drivers, but the deleterious effect of passenger mutations is not yet dominant – this is predicted from theory (McFrand et al, PNAS 2013 and 2014) and empirical evidence from our previous analysis of TCGA data (Persi et al, PNAS, 2018), as described in the

Introduction, and now also presented in revised **Figure 7A**. The revised analysis now further validates this non-monotonic behavior in individual patients (**Figures 5 and 6**).

Although the statistics in the manuscript left me with many unanswered questions, I believe overall that dN/dS values in treated tumors drop as the authors claim. However, **I think the simplest explanation for this phenomenon** – which is also in line with the larger literature – is that treatment causes cancers to go through **bottlenecks**, from which they emerge with more (mostly neutral) mutations that have accumulated during tumor growth. These mutations depress the dN/dS ratio because regular chemotherapy treatment typically does not elicit resistance through specific driver/resistance mutations, and even targeted treatment resistance is usually mediated by a very small set of functional mutations. However, the treatment bottlenecks expose many neutral variants that have accumulated during years and decades of growth. I think that is essentially what the authors are seeing here.

Again, we stress that our principal observation is not a tendency to lower dN/dS values, but rather a shift to neutrality. Furthermore, we devoted an entire section in the **Results** to the issue of what is the possible role of treatment induced bottlenecks (“**Relationship between dN/dS and Tumor size**”) in driving tumors to neutrality – this is indeed a reasonable expectation because therapy would decrease the tumor size which in turn can result in random sampling and thereby genetic drift and neutrality. This contrasts with the driving force we suggest: that neutrality leads to enhanced tumor growth (as the regime of high tumor fitness). To test this, we examined the association of dN/dS with tumor size. The results indicate that the correlation is generally weak, but when significant, tumor size is larger when the regime is close to neutrality in most cases (but not all). This indicates that the effect of neutrality is to facilitate tumor growth although population bottlenecks can be a contributing factor in driving neutrality.

Generally, we agree that many resistant variants are pre-existent, but this is not always the case and therapy can shape the emergence of variants (as explained in detail in the **Introduction**). Our methodology can help in detecting when positive selection occurs in late stages under treatment (see the revised **Figure 6**, and patient-specific signature of the bladder patient P117).

It is universally acknowledged **that most selection happens on the way from normal tissue to the tumor founder cell**, i.e. most driver mutations that would **elevate dN/dS are happening along the tumor trunk**. After that, this driver contingent stays relatively invariant, even during metastasis (see eg Reiter et al. Science 2018) and many people believe that many tumors essentially evolve neutrally after the birth of the founder cell (see e.g. Williams et al, Nature Genetics 2016). **Evidence of subclonal evolution is hard to demonstrate**, it typically requires multi-region sequencing at sufficient depth. So I am not sure what the authors really mean when they say there is a “shift to neutrality” after treatment failure – such a thing cannot be proven by a crude measure like genome-wide dN/dS in my opinion. It would require careful multi-region sampling in treated and untreated tumors to demonstrate that in untreated tumors, evidence of subclonal evolution beyond the variants that are obviously enriched in the tumor trunk is happening, and it would require that the authors show that the same is not true after treatment – a tall order indeed. In the absence of such an analysis, I think that the claims this manuscript makes are not substantiated.

The revised manuscript includes an extensive analysis of the relationship between dN/dS and clonality using 3 levels of analysis (by phylogenetic trees analysis, clonality composition – FISH plots, and the correlation between dN/dS and N as function of VAF) – see details in reply to **point**

7 of **Reviewer 1**. The results indeed show the dominant positive selection in the tumor trunk, but the branches are not neutral and manifest strong negative/purifying selection, such that the overall regime (considering all mutations) is close to neutral – this occurs in all scenarios (untreated and treated cancers and regardless of the tree shape, i.e., with long or short branches) – see revised **Figure 5**. This observation is then further supported by analysis of the correlation between dN/dS and N as function of VAF, showing signatures of positive selection for early mutations vs. signatures of negative selection in late mutation (but an overall dominant neutral evolution) – see revised **Figure 6**. These results overall provide further validation and evidence for our working assumption, based on theory and previous empirical evidence (presented in **Figure 7A**) – this time, in single patients. We hope that this demonstrates that our approach can be a useful additional tool that can be used to assess important aspects of tumor evolution to assist decision making in the clinic – it is different (and easier to apply) than existing methods the Reviewer mentions.

A few more specific comments:

The regression approach that yields p-values throughout the paper is highly unusual and cannot be understood in detail from the methods. It also yields rather suspect results, like a p-value of 0 for the regression line in figure 2B where only a single data point shows a drop in dN/dS while all other data points are clustered together near 1,1. Can the authors replicate their statistics with more standard methods, e.g. a Wilcoxon signed-rank test would work in this scenario?

We are not sure what is unusual about a standard linear regression fit of scatter data in 2 dimensions. Nor do we understand what is unusual about ANOVA F-test which we use for the comparison between two linear regressions. We also do not understand the relevance of Wilcoxon rank test to the analysis presented. The only part of this work where rank tests are relevant is the survival analysis, which we use to assess the P-values in the Kaplan-Meier analysis. All these standard statistical tools are adequately described in **Methods** in our humble opinion. We also stress that the CODE.zip package includes all the (extended) statistical analysis and is fully reproducible in MATLAB.

In general, I find the paper's statistical arguments are weak in places and sometimes not well justified. For example, the text claims "The post-treatment dN/dS values close to neutrality were associated with the worst outcome (rapid relapse), suggesting that the shift to neutrality is associated with higher tumor fitness and treatment failure (Fig. S4)." – but Fig. S4 has no statistical analysis, and the claim of a quadratic relationship between time to relapse and dN/dS is not believable, as it's essentially driven by a single data point. The authors also do not explain why a quadratic relationship would be expected based on any biological considerations.

The analysis of time to relapse is secondary to the survival analysis and only provides further evidence to the claim that tumor fitness is higher around neutrality. In Figure S4, this fit is only qualitative, with the purpose of showing that the time to relapse (in most cases) is shorter around values of dN/dS close to 1. It is not derived by 1 point, rather there is a clear tendency for a decrease (negative slope) when dN/dS < 1 and an increase (positive slope) when dN/dS > 1. Because this fit can be misleading (with the minimum shifted to high values of dN/dS, leading to this confusion), we removed it from Fig S4, and leave this figure as qualitative evidence (similarly to the case of CLL in Fig S5). We believe the point of high tumor fitness is strongly supported and explained in the revised manuscript, as detailed above.

Reviewer #5 (Remarks to the Author): Early Career Researcher co-reviewer

We appreciate this initiative, as well as Nature acknowledging it.

Reviewer #1 (Remarks to the Author):

The authors have satisfactorily answered my questions and meaningfully revised the manuscript. I have no further suggestions or comments.

We thank the Reviewer once again for the constructive feedback and suggestions, and we are glad the Reviewer found our revision meaningful and satisfying.

Reviewer #2 (Remarks to the Author):

The authors have answered my concerns.

Reviewer #2 (Remarks on code availability):

Ok

We thank the Reviewer once again for the constructive feedback and suggestions, and we are glad that our revision answered all the concerns. We also appreciate checking that our code is fine and reproduces the results.

Reviewer #3 (Remarks to the Author):

We appreciate this initiative and thank again the Reviewer for participating in the review.

Reviewer #4 (Remarks to the Author):

I regret to say that this rebuttal and revision did not significantly change my assessment of the manuscript, primarily because I did not feel that the authors engaged in a good faith effort to answer the raised points. Most importantly, they failed to address our primary concern that dN/dS may be confounded by mutation burden. As we explained, we expect that post-treatment samples have a substantially greater (passenger) mutation burden than pre-treatment samples, raising the concern that signals of selection are 'drowned out' by non-functional mutations to a greater degree post-treatment. The authors did not perform analyses to alleviate this central concern, nor did they directly respond to it. Instead, they turned to an analysis of tumor size which is of questionable relevance given the uncertain correlation between tumor size and mutation burden. I note that no direct replies or analyses were produced in the rebuttal, and the revised manuscript did not highlight revisions, it is therefore largely unclear which changes – if any – the authors may have made in response to the review. Similarly, the concern that a lot of the results in the paper were driven by single or extremely limited data points (a real issue in Figures 2 and 3) was dismissed by the authors.

First and foremost, we regret that our reply and revised manuscript left a negative impression and would like to assure the reviewer(s) that we attempted, fully in good faith, to address thoroughly every single point raised by all the reviewers. We believe our revision was substantial, included numerous new analyses (that took significant effort and time) and we have highlighted throughout our reply all the changes to the manuscript (by bolding “in the revised manuscript...” as also done below). Most importantly, we broke down the Reviewer previous feedback, replied to each comment and clarified each argument, including pointing out that we do not observe a systematic decrease in dN/dS post-treatment (as claimed by the Reviewer), and evaluating the dynamic of dN/dS along the phylogenetic tree of individual patients (cf. Figure 5A and SI) answering the main questions regarding positive selection at the trunk and balancing/negative selection in later phases. Perhaps, simply due to the numerical order of the Reviewers, we sometimes referred to the replies and points of previous Reviewers (to avoid producing an excessively repetitive and lengthy response). In doing so, our sole goal was and remains to produce an improved analysis with the hope that it will serve the scientific community (and eventually contribute to human health), and we always welcome feedback to allow us to improve accordingly, constructively and as appropriate.

Regarding the specific points raised again by the Reviewer. We fully recognize and understand the possible effect that mutation burden, in theory, might have on dN/dS estimates. However, as we explain below, it is unlikely to be a significant concern in the present analysis and hardly could affect the results obtained. Specifically, bias in dN/dS due to mutation burden may occur either due to (weak) selection affecting synonymous sites [e.g., Rahman et al, 2021], or due to the saturation of dS over long evolutionary times. Both scenarios would result in artifactually low dS values and accordingly to inflated dN/dS values (except for a very strong positive selection regime; then, dN may saturate before dS), that is, underestimation of purifying selection and/or overestimation of positive selection. **However**, the first scenario (weak selection) generally results in a mild bias of dN/dS and is unlikely to be relevant in cancer evolution because the vast majority of variants appear to be neutral [see Martincorena et al 2017, Weghorn & Sunyaev 2017, Persi et al, 2018, all referred to in the current manuscript]. The second scenario (saturated synonymous sites) hardly could be relevant because it is unlikely that during tumor evolution a site mutates more than once, especially in low mutation burden cancers (many of which are included in our analysis). For this reason and because we analyzed the entire genome (i.e., very long and not highly diverged sequences), we did not employ long-evolutionary time corrections in our dN/dS estimates. Such corrections might be more relevant when evaluating dN/dS in single genes (e.g., see Martincorena et al, 2017). Further, our estimates of dN/dS (dominant neutrality and deviations from it) are generally in agreement with many other studies (see Persi et al, 2018), and possible biases in dN/dS estimates were assessed in various ways, including by fitting mutation data to a neutral model, and experimentally, with respect to various factors - as detailed in **Methods**.

Furthermore, several points in our analysis further indicate that the mutation burden is not a significant confounding factor in our analysis. **First**, even if the bias in dN/dS existed and was significant, it would lead to a systematic shift of the linear regression fit post-treatment, rather than to rotation of the regression line relative to the pre-treatment regression. This rotation reflects the trend toward neutrality in tumors resisting treatment, which is our major observation, and the association of the shift to neutrality with worse prognosis post-treatment is well demonstrated.

Thus, this potential bias does not appear to be reflected in our results. **Second**, and most importantly, our observation of the rotation, reflecting a tendency to neutrality post-treatment is quite universal and applies both to low-burden cancers (e.g., ALL) and to mid/high-burden cancers (e.g., glioblastoma). Thus, the main results of this study do not seem to be confounded by the mutation burden status. **Third**, the expectation of “substantially greater (passenger) mutation burden” post-treatment may indeed reflect the typical case, but it is not always the case. Similarly, the assumption that passengers are functionally irrelevant is not necessarily valid because many passengers are likely to be, at least, slightly deleterious with respect to the tumor fitness. In our analysis, even when the numbers of mutations pre vs. post treatment are comparable (that is, no substantial strong increase post-treatment), we observe the shift to neutrality (e.g., MM in Fig. 4). In contrast, in other cases, where the increase in the number of mutations is apparent (cf. Fig 6C; a bladder cancer patient), we do not necessarily observe a decrease in dN/dS (in this case, dN/dS continues to increase, reflecting continued accumulation of drivers post treatment). Thus, we do not obtain a systematic bias in dN/dS in high burden, rather the behavior depends more critically on the accumulation of drivers (and the balance with passengers), which differ from patient to patient and could be affected by treatment. Once again, the mutation burden does not seem to confound our results – dN/dS can go both ways and our analysis explains why it happens and how it can assist to interpret and guide treatment. **Lastly**, we have to note that the analysis with respect to tumor size was reported already in the initial submission, not as a reply to the Reviewer, and it served to address a completely different point, namely, the potential contribution of tumor shrinkage (and sampling bias) to neutrality, which we attempted to clarify. **In the revised manuscript**, we now emphasize in the **Methods** (subsection “Formula” under “Evaluating the Evolutionary State of Tumor Genomes by N and dN/dS) that the dN/dS metric assumes that synonymous mutations are neutral and unsaturated. Further, in the last paragraph of the **Discussion** that lists future directions, we added that effects of possible biases in dN/dS (such as selected or saturated synonymous mutations) should be considered in future studies, especially in high-burden cases (to better assess the true strength of negative selection).

We also recognize throughout the current manuscript that the analysis is based on relatively small numbers of patients (and samples per patient), and this is the reason for designing the study in such a manner as to analyze as many cohorts as we could possibly find, and for validating the results on published and original data. This is simply the unfortunate reality of the current state of the art (that is, the number of biopsies taken from a patient is typically small low, being minimized for understandable reasons, and many relevant studies do not provide data on synonymous mutations due to the clinical focus on ‘drivers’). Again, this is one reason why we added the analysis of original data (MM) which contains a substantial number of patients, with samples tracked over time and during a line of treatments. **In the revised manuscript**, we now added in the last part of the discussion that future studies should continue to evaluate the current results as cancer-specific or case-specific deviation may exist and it would be important to understand why.

Reviewer #5 (Remarks to the Author):

We appreciate this initiative and thank again the Reviewer for participating in the review.